# Cell state dependent effects of Bmal1 on melanoma immunity and tumorigenicity

Xue Zhang [1,2,3] ✉, Shishir M. Pant[4,5,6], Cecily C. Ritch [4,5,7], Hsin-Yao Tang [1], Hongguang Shao[1], Harsh Dweep[1], Yao-Yu Gong [1,2], Rebekah Brooks[1,2], Patricia Brafford[1,2], Adam J. Wolpaw [1,8,9], Yool Lee [10], Ashani Weeraratna[11], Amita Sehgal [12], Meenhard Herlyn [1], Andrew Kossenkov[1], David Speicher[1], Peter K. Sorger [4,5,6], Sandro Santagata [4,5,6,7] & Chi V. Dang [1,2,3,11] ✉

The circadian clock regulator Bmal1 modulates tumorigenesis, but its reported effects are inconsistent. Here, we show that Bmal1 has a context-dependent role in mouse melanoma tumor growth. Loss of Bmal1 in YUMM2.1 or B16-F10 melanoma cells eliminates clock function and diminishes hypoxic gene expression and tumorigenesis, which could be rescued by ectopic expression of HIF1α in YUMM2.1 cells. By contrast, over-expressed wild-type or a transcriptionally inactive mutant Bmal1 non-canonically sequester myosin heavy chain 9 (Myh9) to increase MRTF-SRF activity and AP-1 transcriptional signature, and shift YUMM2.1 cells from a Sox10[high] to a Sox9[high] immune resistant, mesenchymal cell state that is found in human melanomas. Our work describes a link between Bmal1, Myh9, mouse melanoma cell plasticity, and tumor immunity. This connection may underlie cancer therapeutic resistance and underpin the link between the circadian clock, MRTF-SRF and the cytoskeleton.

Circadian biological rhythms are coupled organismal and cellular activities that oscillate with a period of 24 h in synchrony with the day-night solar cycle. These cycles allow organisms to anticipate food availability and daily periods of sleep[1–3], enabling fitness and longevity[4]. Clock disruption in humans or mice result in obesity, inflammation, and predisposition to cancer development and progression[5–7]. Further, analyses of data from The Cancer Genome Atlas (TCGA) suggest that human cancers have genomic alterations of clock regulators[8,9], some of which participate in tumorigenesis in mice[7,10,11].

Here, we focus on whether the cell autonomous circadian clock affects melanoma tumorigenesis. Melanocytes, which are derived from Sox9-driven neural crest cells[12], acquire oncogenic mutations that give rise to melanoma. Human melanomas display substantial phenotypic heterogeneity and variation in markers with SOX10 being commonly expressed[13]. However, melanoma cell states can be variable including cells that are heavily pigmented and well-differentiated (SOX10[high]/ MITF[high]), neural crest-like (NGFR[high]), or undifferentiated and drug resistant (SOX9[high]/AXL[high])[14]. Intriguingly, melanomas under therapeutic stress imposed by BRAF inhibitors display lineage plasticity

[1]The Wistar Institute, Philadelphia, PA, USA. [2]Ludwig Institute for Cancer Research, New York, NY, USA. [3]Bloomberg-Kimmel Institute for Cancer Immunotherapy, Department of Oncology, Johns Hopkins University School of Medicine, Baltimore, MD, USA. [4]Laboratory of Systems Pharmacology, Harvard Medical School, Boston, MA, USA. [5]Ludwig Center at Harvard, Harvard Medical School, Boston, MA, USA. [6]Department of Systems Biology, Harvard Medical School, Boston, MA, USA. [7]Department of Pathology, Brigham and Women's Hospital, Harvard Medical School, Boston, MA, USA. [8]Division of Oncology, Department of Pediatrics, Children's Hospital of Philadelphia, Philadelphia, PA, USA. [9]Center for Childhood Research, Children's Hospital of Philadelphia, Philadelphia, PA, USA. [10]Department of Translational Medicine and Physiology, Elson S. Floyd College of Medicine, Washington State University, Spokane, WA, USA. [11]Department of Biochemistry and Molecular Biology, Bloomberg School of Public Health, Johns Hopkins University, Baltimore, MD, USA. [12]Howard Hughes Medical Institute, Chronobiology and Sleep Institute, Perelman School of Medicine, University of Pennsylvania, Philadelphia, PA, USA. ✉e-mail: xzhan325@jh.edu; cvdang@jhmi.edu

with transient expression of NGFR, EGFR, AXL, and an AP-1 transcriptional signature and with loss of SOX10[15–17]. However, whether the circadian clock affects melanoma cell plasticity is unknown[13]. As such, we explore here the effects of the circadian regulator Bmal1 on mouse melanoma cell state, tumorigenesis and therapeutic resistance.

## Results

### Diminished tumorigenesis of murine Bmal1-null YUMM2.1 melanoma rescued by Hif1α

We previously reported that loss of Bmal1 (Bmal1KO) in the B16-F10 murine melanoma cell line diminished tumor growth in immunocompetent C57BL/6 mice and altered time-of-day specificity of response to anti-cancer drugs[18]. To provide insight into Bmal1's effects on B16-F10 tumor growth, we used QuantSeq 3' mRNA sequencing to identify differentially expressed genes (DEGs) between control (Ctrl) and Bmal1-null in B16-F10 (Fig. 1a and Supplementary Data 1). Gene set enrichment analysis (GSEA) reveals an enrichment of hypoxia responsive genes in Ctrl cells (Fig. 1b and Supplementary Fig. 1a). Congruently, hypoxic B16-F10 Ctrl cells compared to Bmal1 KO B16-F10 cells had higher levels of Hif1α and Sox9, a cell state determining factor (Fig. 1c). Notably, Hif1α or Sox9 expression in melanoma is associated with tumorigenesis[19,20], suggesting their putative roles in Bmal1-dependent B16-F10 tumorigenesis.

To further probe the role of Bmal1 in melanoma, we studied another mouse melanoma cell line YUMM2.1, which is derived from transgenic tumors driven by human melanoma-relevant mutations Braf$^{V600E}$, Pten$^{-/-}$, Cdkn2a$^{+/-}$, and Bcat$^{STA/+}$ (heterozygous for a stabilized mutant β-catenin)[21,22]. We knocked out Bmal1 in YUMM2.1 and studied three Bmal1-null clones (aC3, aG9 and cD6) and three control clones (B8, C8 and F5) (Fig. 1d and Supplementary Fig. 1b). We found that Bmal1-null clones lost clock function as determined by the circadian Bmal1-promoter-Luciferase reporter (Bmal1::dLUC) (Fig. 1e and Supplementary Fig. 1c), and that these clones collectively had lower levels of Hif1α, Sox9 (Fig. 1d and Supplementary Fig. 1b) and diminished tumorigenesis in C57BL/6 mice (Fig. 1f and Supplementary Fig. 1d), consistent with the effect seen with B16-F10 cells.

We then performed genetic complementation experiments with the aC3 clone YUMM2.1 Bmal1-null (YUMM2.1KO) cells to determine whether the effects of loss of Bmal1 could be reversed by constitutive expression of (1) wild-type (WT) Bmal1 (YUMM2.1KO-WT), (2) a transcriptionally inactive Bmal1 mutant lacking the basic helix-loop-helix DNA binding domain (dHLH) (YUMM2.1KO-dHLH) or (3) empty vector (YUMM2.1KO-EV) (Fig. 1g). dHLH-Bmal1 can dimerize with Clock and inhibits endogenous Clock-Bmal1 transcriptional activity but retains translational stimulating activity[23]. As expected from previous studies[24–26], reconstitution of WT-Bmal1, but not dHLH-Bmal1 or EV, reestablished clock function (Supplementary Fig. 1e). RNA-sequencing data showed that comparing with EV and dHLH-Bmal1, complementation with WT-Bmal1 altered the expression of many genes (Fig. 1h and Supplementary Data 1) which included Bmal1-Clock targets (e.g., Dbp, Nr1d1, Nr1d2, Per2, and Per3) (Fig. 1i) and Sox9 (Fig. 1j).

Congruent with observations in B16-F10 cells (Fig. 1b), hypoxia responsive genes were enriched by complementation by WT-Bmal1 (Fig. 1k and Supplementary Fig. 1f) but not by dHLH-Bmal1 (Supplementary Fig. 1g). Further, YUMM2.1KO-WT cells had higher sustained Hif1α and Sox9 protein levels under hypoxia than YUMM2.1KO-EV or YUMM2.1KO-dHLH (Fig. 1g). Despite these expression differences, we found that their growth rates were indistinguishable in vitro under normoxia (Supplementary Fig. 1h) or hypoxia (Supplementary Fig. 1i). However, WT-Bmal1, in contrast to EV or dHLH-Bmal1, could increase tumor growth (Fig. 1l) in vivo, suggesting that Bmal1 may augment tumorigenesis by stabilizing Hif1α protein[27] rather than Hif1α mRNA which was not induced (Supplementary Fig. 1j). In this regard, we ectopically expressed a stabilized Hif1α-TM allele in YUMM2.1

(Supplementary Fig. 1k) and YUMM2.1KO cells (Fig. 1m) and found that Hif1α-TM increased tumorigenesis of YUMM2.1 KO cells (Fig. 1n, o) but not significantly in YUMM2.1 cells (Supplementary Fig. 1k, l) when compared to control. We could not stably express Hif1α-TM in Bmal1 KO B16-F10 cells, hence the role of Hif1α in Bmal1 KO B16-F10 tumorigenesis is unclear. Nonetheless, these observations collectively suggest that loss of Bmal1 diminishes tumorigenesis at least partly through reduction in Hif1α activity.

### Ectopic expression of Bmal1 affects immune infiltration and increases YUMM2.1 tumorigenesis

Having observed that loss of endogenous Bmal1 diminishes tumorigenesis, we then sought to determine how ectopic expression of Bmal1 or dHLH-Bmal1 affects tumorigenesis in the native YUMM2.1 cell state. It is notable that ectopic Bmal1 expression in the native background have potential neomorphic effects[28,29], such as the recently reported association of deregulated BMAL1 expression with anti-androgen therapy resistance in human prostate cancer[30]. We generated YUMM2.1 empty vector (YUMM2.1-EV), dHLH-Bmal1 (YUMM2.1-dHLH) and WT-Bmal1 (YUMM2.1-WT) expressing YUMM2.1 cell lines (Fig. 2a). Unexpectedly, although Hif1α was clearly diminished in Bmal1 KO cells, it was not consistently increased in YUMM2.1-WT or YUMM2.1-dHLH cells across the time course (Fig. 2a). Whereas YUMM2.1-EV and YUMM2.1-WT cells retained circadian clock function as measured using the Bmal1::dLUC reporter, YUMM2.1-dHLH cells had disrupted clock function (Supplementary Fig. 2a) attributed to the dominant negative transcriptional effect of dHLH-Bmal1[23] which suppressed the expression of core Bmal1 targets, Dbp, Nr1d1, and Nr1d2 (Supplementary Fig. 2b).

In contrast to the indistinguishable in vitro growth rates of these cell lines (Supplementary Fig. 2c), the modest increase of ectopic WT-Bmal1 protein expression in YUMM2.1-WT significantly increased tumor growth as compared to YUMM2.1-EV tumors (Fig. 2b) in C57BL/6 mice. However, tumor growth was indistinguishable in immunocompromised NSG mice (Fig. 2c), suggesting the host immune system plays a role in the difference in tumorigenesis. To explore this further, we performed cytokine measurements on supernatant and found that WT-Bmal1 and dHLH-Bmal1 cells grown in vitro displayed higher levels of VEGF, GM-CSF, Igfbp-6, Il-23, KC (Cxcl1), and M-CSF compared with the supernatants of EV cells (Fig. 2d and Supplementary Fig. 2d). Further, tumor immunophenotyping by flow cytometry (gating strategy: Supplementary Fig. 2i) revealed that YUMM2.1-dHLH tumors had significant decrease in Cd103+ dendritic cells and NK cells and a notable increase in M-MDSCs, M2 macrophages (Cd11b$^+$; F4/80$^+$; Cd11c$^-$; Fig. 2e) compared to YUMM2.1-EV or YUMM2.1-WT tumors. The influx of myeloid cells in YUMM2.1-dHLH tumors correlates with high levels of GM-CSF and M-CSF secreted by these tumor cells compared to cells expressing WT-Bmal1 or EV (Fig. 2d). Both WT-Bmal1 and dHLH-Bmal1 tumors had increased PMN-MDSCs and Cd4+ T cells compared to EV tumors (Fig. 2e).

The changes in tumor immune infiltration suggest that dHLH-Bmal1, more than WT-Bmal1, confers an immunosuppressive tumor microenvironment. To test this hypothesis, we treated comparable sized (~100 mm³) YUMM2.1 tumors with anti-PD1 (Fig. 2f). YUMM2.1-WT and YUMM2.1-dHLH tumors, which have increased PMN-MDSCs and Cd4+ T cells (Fig. 2e), were more resistant to anti-PD1 treatment compared to YUMM2.1-EV tumors (Fig. 2f). By comparing anti-PD1 treated tumors, it is notable that YUMM2.1-EV tumors were smallest, whereas YUMM2.1-WT and YUMM2.1-dHLH tumors grew to larger sizes (Supplementary Fig. 2e). These responses replicated results from an experiment (Supplementary Fig. 2f), wherein some of the YUMM2.1-EV tumors treated at a smaller size regressed with anti-PD1 treatment, while YUMM2.1-dHLH were most resistant. We note that YUMM2.1-WT

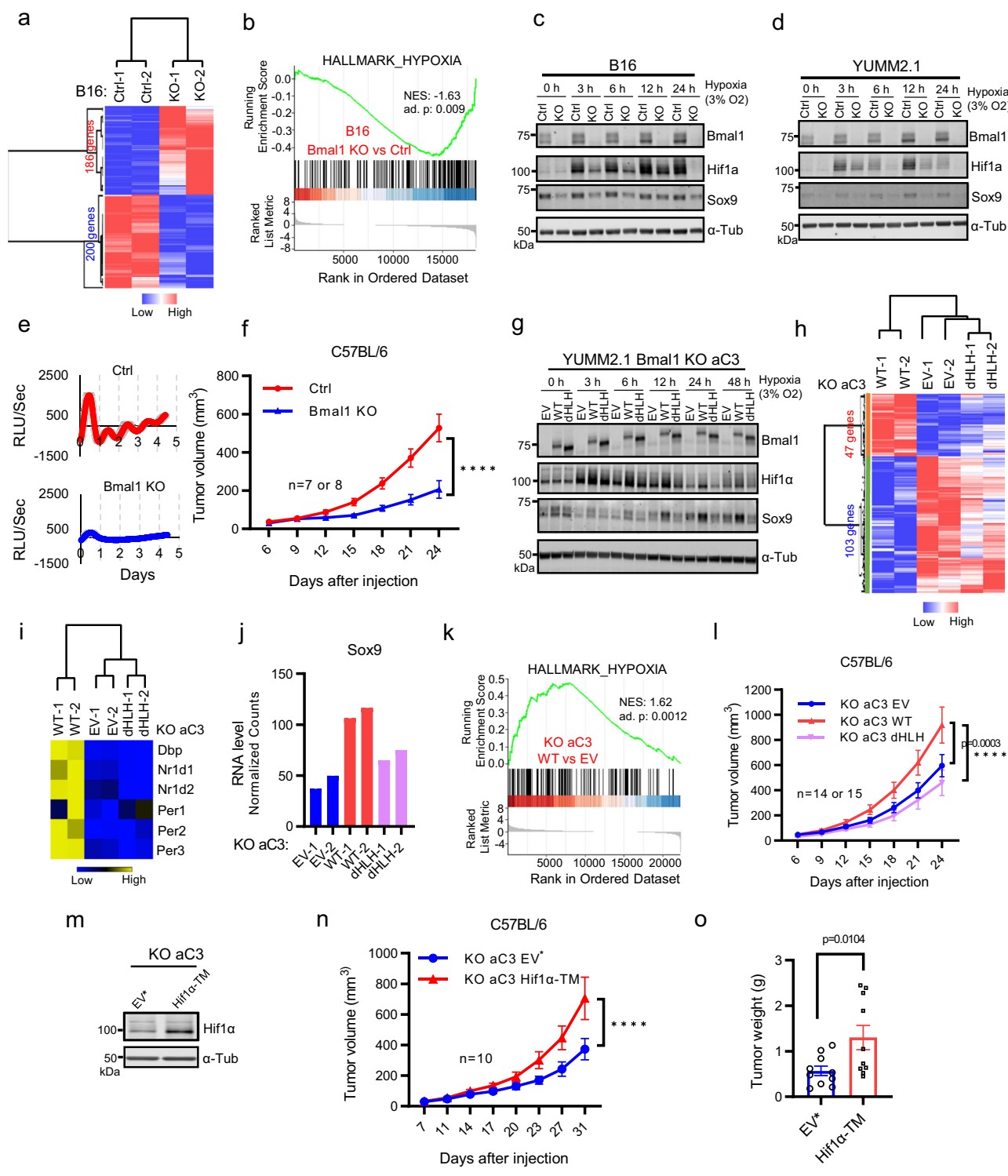

tumors grew faster than the more anti-PD1 resistant YUMM2.1-dHLH tumors. Correlating with slower tumor growth rates of YUMM2.1-dHLH tumors, the percentage of Ki67+ melanoma cells (Sox9+ or Sox10+ cells) were lower in YUMM2.1-dHLH compared to YUMM2.1-WT tumors (Supplementary Fig. 2g, h). In this respect, despite being more resistant to anti-PD1 therapy, the proliferative fraction of YUMM2.1-dHLH tumors was less than that of YUMM2.1-WT tumors and may account for the difference in overall tumor growth rates. Taken together, our findings indicate that ectopic expression of Bmal1 proteins in the YUMM2.1 native state confers immune resistance and altered tumor growth.

## Ectopic expression of Bmal1 shifts YUMM2.1 cells toward a mesenchymal state

To further explore the factors contributing to the differences in immune infiltration and immunotherapy resistance induced by dHLH-Bmal1 and WT-Bmal1 versus EV, we analyzed the transcriptomes of the YUMM2.1 cell lines and identified DEGs (Fig. 3a, Supplementary Fig. 3a, and Supplementary Data 2). DEGs did not reveal specific clues that explain the higher Ki67+ fraction and growth rates of WT-Bmal1 expressing tumors compared to those expressing dHLH-Bmal1, which profoundly suppressed the core circadian clock (Supplementary Fig. 2b). Intriguingly, however, GSEA revealed epithelial to

**Fig. 1 | Loss of Bmal1 decreases YUMM2.1 tumorigenesis. a** Heatmap for DEGs in B16 Control (Ctrl) vs B16 Bmal1-null (KO) cells. BR of 2. **b** GSEA showing hypoxia gene set significantly enriched in B16 Control vs Bmal1 KO. **c, d** Immunoblot of Bmal1, Hifα and Sox9 proteins in Ctrl and Bmal1 KO clones from B16 cells (**c**) or YUMM2.1 cells (**d**) at different time points after exposure to 3% O$_2$. α-Tubulin served as loading control for all immunoblots except where noted. RE of 3 for **c** and 2 for **d. e** Real-time luminescence monitoring of *Bmal1*::dLUC in YUMM2.1 Bmal1 KO clone aC3 and control clone B8 synchronized with dexamethasone for up to 4.5 days. Luminescence signal is baseline subtracted and data are shown starting 24 h after synchronization. Signal confidence interval is shown in gray. Mean ± SEM of BR of 3, RE of 2. **f** Tumor growth rate of B8 (*n* = 7) and aC3 clones in male C57BL/6 mice (*n* = 8). Mean ± SEM; ****$P$ < 0.0001 by Two-way ANOVA test. **g** Immunoblot for Bmal1, Hifα and Sox9 protein levels in Bmal1 KO aC3 clone with empty vector (EV), WT-Bmal1 and dHLH-Bmal1. RE of 3. **h**–**k** Quant-seq data

analyses from aC3 clone with EV, WT-Bmal1 and dHLH-Bmal1. Samples are in duplicates. Heatmap for all 47 increased and 103 decreased genes by WT-Bmal1 but not dHLH-Bmal1 vs EV (**h**); Heatmap for relative expression of Bmal1 direct target genes (**i**); Normalized counts of Sox9 (**j**) and GSEA showing the enrichment of hypoxia gene set with WT-Bmal1 vs EV cells (**k**). **l** Tumor growth rate of aC3 with EV (*n* = 14), WT-Bmal1 (*n* = 15) and dHLH-Bmal1 (*n* = 14) in male C57BL/6 mice. Mean ± SEM, RE of 2. ****$p$ < 0.0001 by Two-way ANOVA test followed by Tukey's multiple comparisons test. **m** Immunoblot for Hifα in aC3 clone with EV* and Hifα-TM. Note: aC3 EV* is different from aC3 EV which has *Bmal1*::dLuc reporter. RE of 2. Tumor growth rate (**n**) and tumor weight (**o**) of aC3 with EV* and Hifα-TM in male C57BL/6 mice (*n* = 10 in each group). Mean ± SEM. *p*-value by Two-way ANOVA test in **n**. Mean ± SEM. Two-tailed *p*-value by unpaired *t*-test in **o**. RE of 2. BR = biological replicate, RE = replicate experiment. Source data are provided as a Source Data file.

mesenchymal transition (EMT) gene signature was enriched in both YUMM2.1-WT and YUMM2.1-dHLH versus YUMM2.1 EV cells (Fig. 3b, c). Ectopic WT-Bmal1 and dHLH-Bmal1 increased genes associated with mesenchymal melanoma state, such as Sox9, Fn1, Col1a1, Prrx2, Klf4, Snai2, Twist2, Lmo1 and Axl[31] (Fig. 3a). Notably, expression of dHLH-Bmal1, more so than WT-Bmal1, increased Sox9 expression and repressed Sox10 (Fig. 3d), whereas WT-Bmal1 but not dHLH-Bmal1 increased Sox9 expression in Bmal1-null YUMM2.1 cells (Fig. 1j). Although hypoxia gene signature was enriched by ectopic dHLH-Bmal1 versus EV, it was not enriched by WT-Bmal1 (Supplementary Fig. 3b, c). These differences in how Bmal1 and dHLH-Bmal1 expression impact Sox9 and hypoxia response in Bmal1-null vs Bmal1-WT cells demonstrates an important context-dependent effect of perturbing the molecular circadian clock.

We then sought to determine potential drivers of Sox9 expression and mesenchymal transition by determining consensus transcription factor motif enrichment in DEGs driven by WT-Bmal1 and dHLH-Bmal1. The consensus motif for the AP-1 (Fig. 3e), which has been documented to drive melanoma mesenchymal state[32,33], was significantly enriched in promoter regions (<3 kb TSS) of DEGs. Expression of several AP-1 factors are elevated by WT-Bmal1 and dHLH-Bmal1 (Fig. 3f, g). In addition, chromatin states driven by WT-Bmal1 and dHLH-Bmal1 were determined by chromatin immunoprecipitation sequencing (ChIP-seq) for H3K4me3, H3K27me3, and H3K27ac. The 542 top DEGs had increased H3K4me3 and H3K27ac within 3 kb of the transcription start site (TSS), whereas H3K27me3 was diminished (Supplementary Fig. 3d and Supplementary Data 3, 4). Further analysis of H3K27ac, an enhancer activation marker, revealed 1187 regions overlapping between WT-Bmal1 and dHLH-Bmal1, acquired this modification across the genome (Fig. 3h). The AP-1 motif is significantly prevalent in these regions (Fig. 3i), including the 131 regions that are intergenic (>20 kb from TSS) (Fig. 3j and Supplementary Fig. 3e). Notably, dHLH-Bmal1, more so than WT-Bmal1, increased H3K4me3 and H3K27ac at Sox9 and conversely decreased these histone markers at Sox10 (Fig. 3k). Correspondingly, H3K27me3 decreased at Sox9 and increased at Sox10 (Supplementary Fig. 3f). These changes are consistent with increased Sox9 and decreased Sox10 expression in WT-Bmal1 and particularly dHLH-Bmal1 expressing cells in vitro (Fig. 3d and Supplementary Fig. 3g) and in tumors in situ as shown by multiplexed tissue imaging using cyclic immunofluorescence microscopy (CyCIF)[34] (Fig. 3l, m). These findings indicate that ectopic dHLH-Bmal1, more than WT-Bmal1, increased Sox9 expression and shifted YUMM2.1 cells toward a more mesenchymal epigenetic state in mouse melanoma cells enriched with AP-1 motifs in DEGs and enhancers. We found that AP-1 factors, particularly c-JUN and JUNB were also induced by WT-Bmal1 or dHLH-Bmal1 in the human WM3629 melanoma cells with corresponding increase in SOX9 and decrease in SOX10 levels (Supplementary Fig. 3h). Further, by analyzing published scRNAseq data, we identified SOX10$^{low}$/SOX9$^{high}$ human melanomas (Supplementary Fig. 3i) and found that these were immune resistant and characterized

by low HLA-A expression (Supplementary Fig. 3i, j)[35]. Thus, our mouse melanoma model reveals an immune resistant, ectopic Bmal1-driven AP-1-associated Sox9$^{high}$ mesenchymal state that is found in immunotherapy resistant human melanomas.

## Bmal1 interacts with Myh9 in nucleus

The ability of the transcriptionally inactive dHLH-Bmal1 to induce an AP-1 associated mesenchymal melanoma state raises the possibility of a post-transcriptional mechanism. As such, we surmised that the Bmal1 protein interactome in YUMM2.1 cells could reveal a mechanism of action.

We used the TurboID proximity labeling system[36] to identify the Bmal1 interactome. TurboID (Tb) is a promiscuous BirA biotin-protein ligase that is fused to a protein of interest, such as WT-Bmal1 (TbWT) or dHLH-Bmal1 (TbdHLH) (Fig. 4a). Proteins proximal to Bmal1 are biotinylated by the fused Tb moiety and subsequently isolated on streptavidin beads for identification by mass spectrometry. HA-tagged TbWT and TbdHLH fusion proteins were stably expressed in Bmal1 knock-out line YUMM2.1KO aC3, producing YUMM2.1KO-TbWT or YUMM2.1KO-TbdHLH, respectively (Fig. 4a). As controls, we also stably expressed the HA-tagged TurboID without (Tb) or with (TbNLS) a nuclear localization signal in YUMM2.1KO aC3 cells (Fig. 4a). By labeling with biotin for 2 h and using Clock as a positive control (Fig. 4a), we identified proteins that were highly and specifically labeled by TbWT and TbdHLH versus negative Tb or TbNLS controls (Fig. 4b). Notably, Tb and TbNLS non-specifically labeled many cytoplasmic and nuclear proteins, respectively (Fig. 4b and Supplementary Data 5), but these proteins were largely not labeled by either TbWT or TbdHLH proteins, validating the specificity of this labeling technique.

Both TbWT and TbdHLH immunoprecipitated (Fig. 4a; IP: HA) and labeled (Fig. 4c) endogenous Clock, which serves as a positive control for these two fusion proteins. However, unexpectedly, they both highly labeled myosin heavy chain 9 (Myh9) and Actinin 4 (Actn4) whereas other abundant proteins such as Gapdh were poorly labeled (Fig. 4c). Minimal amounts of either Myh9 or Actn4 were labeled by the negative controls TbNLS or Tb (Fig. 4c). These observations suggest that Myh9 labeling by TbWT and TbHLH was non-random, although the interaction between Bmal1 and Myh9 may be transient or low affinity[37,38] as compared to Bmal1-Clock binding. Among the Bmal1 interactors, we chose to focus on Myh9 (non-muscle myosin IIA) in depth, because it has been implicated in melanoma tumorigenesis and drug resistance[39,40], and thought to be a tumor suppressor in several tumor models[41–44].

Myh9 exists as monomers that unfold and assemble into Myh9 multimers to drive cellular movement by ratchetting on actin filaments[45]. Intriguingly, Myh9 is also found in the nucleus (Supplementary Fig. 4a) where it associates with Bmal1[46] and is implicated in regulating transcription[47,48]. To corroborate the interaction of Bmal1 with Myh9, we performed chemical crosslinking to capture low affinity interactions followed by co-immunoprecipitation and found that WT-

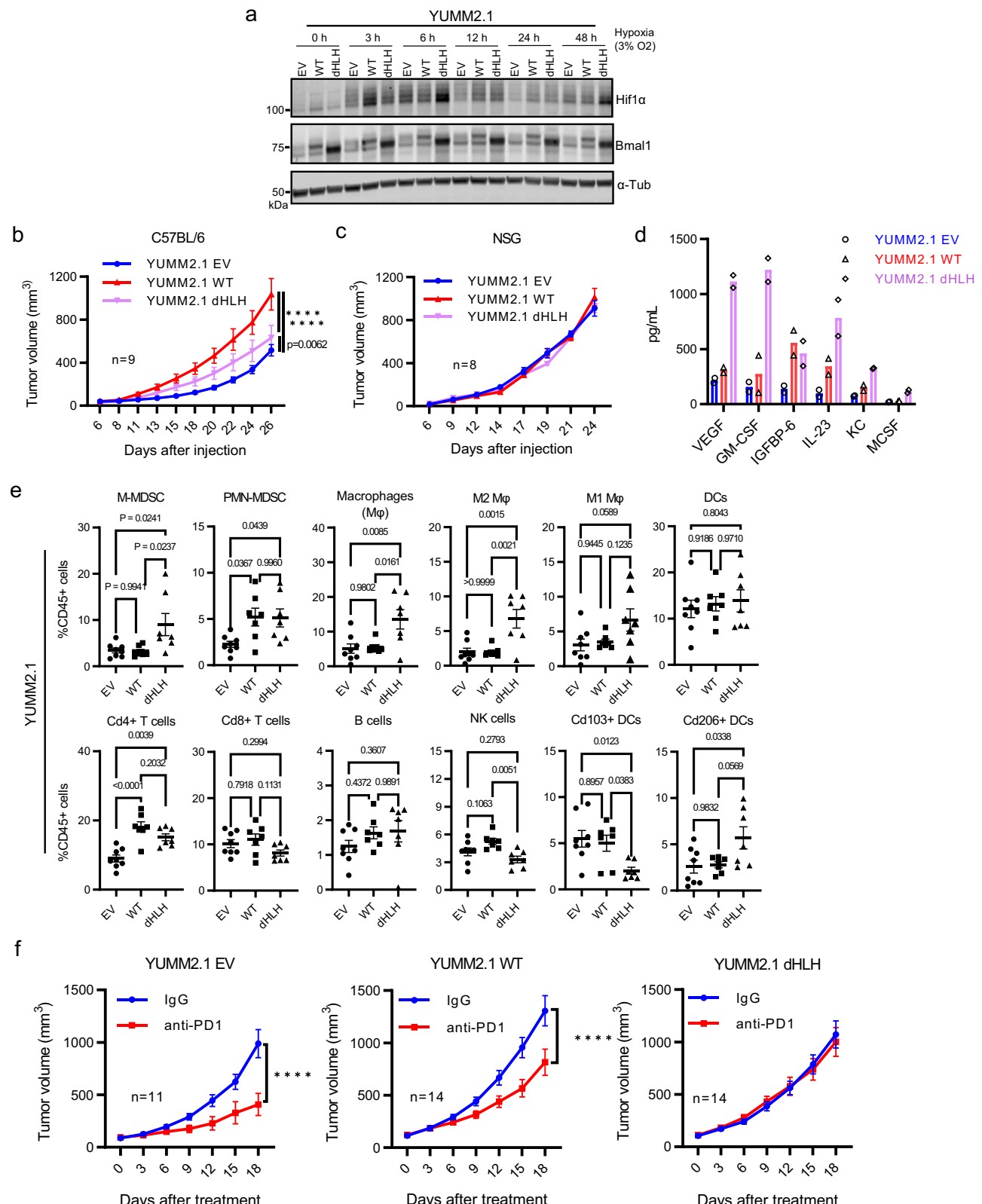

Bmal1 or dHLH-Bmal1 associates with Myh9 and actinin 4 (Actn4), but not with Ezh2, as a control nuclear protein (Fig. 4d). We further mapped the interaction of Bmal1 to the head domain of Myh9 using an overexpression system in 293 T cells (Fig. 4e). To demonstrate the potential in situ association between Bmal1 and Myh9, proximity ligation assay (PLA) was performed, revealing that endogenous or

ectopically expressed Bmal1 proteins are proximal to Myh9, mostly in the nucleus of mouse YUMM2.1 and several human melanoma cell lines (Fig. 4f, g and Supplementary Fig. 4b,c). In addition, the PLA signals between Bmal1 and Myh9 were monotonically increased by WT and dHLH versus EV (Fig. 4g and Supplementary Fig. 4e), which is consistent with dHLH cells being more mesenchymal and resistant to

**Fig. 2 | Ectopic Expression of Bmal1 Affects Immune Infiltration and Accelerates Tumorigenesis of YUMM2.1. a** Immunoblot for Hif1α and Bmal1 in YUMM2.1 with EV, WT-Bmal1 and dHLH-Bmal1. RE of 2. **b, c** Tumor growth rates of YUMM2.1 EV, WT-Bmal1 and dHLH-Bmal1 in male C57BL/6 mice (**b**; $n = 9$) or NSG mice (**c**; $n = 8$). Mean ± SEM. RE of 3 in **b**. RE of 2 in **c**. ****$p < 0.0001$ by Two-way ANOVA test followed by Tukey's multiple comparisons test. **d** The level of cytokines secreted from YUMM2.1 EV, WT-Bmal1 and dHLH-Bmal1 cells as determined by immunoassay (RayBiotech, Mouse Cytokine Array 1000, QAM-CYT-1000). BR of 2.

Each dot represents one biologically independent sample. **e** Flow cytometric immunophenotyping of YUMM2.1 EV ($n = 8$), WT-Bmal1 ($n = 7$) and dHLH-Bmal1 ($n = 7$) tumors in C57BL/6 mice. Mean ± SEM. $p$-value by one-way ANOVA test followed by multiple comparison test. **f** Response of YUMM2.1 EV ($n = 11$ each treatment), WT-Bmal1 ($n = 14$ each treatment) and dHLH-Bmal1 ($n = 14$ each treatment) tumors in male C57BL/6 mice to control IgG or anti-PD1 treatment given IP every 3 days. Mean ± SEM. ****$p < 0.0001$ by Two-way ANOVA test. BR = biological replicate, RE = replicate experiment. Source data are provided as a Source Data file.

immunotherapy. Further, although reconstituted WT or dHLH-Bmal1 interacted with Myh9 in YUMM2.1KO cells, PLA signals were equivalent to signals in YUMM2.1 EV and lower compared to YUMM2.1 WT and dHLH cells (Supplementary Fig. 4d,e). Clock served as a positive control (Supplementary Fig. 4b, c). As additional controls, HA-tagged TbNLS and Tb showed very low PLA signals with Myh9, whereas TbNLS-mKi67 and Tb-filamin A PLA signals were higher (Supplementary Fig. 4b, c) as expected for non-specific nuclear and cytoplasmic labeling, respectively (Supplementary Data 5). Collectively, the evidence supports an interaction between Bmal1 proteins and Myh9 in mouse and human melanoma cell lines.

### Bmal1 and Myh9 interaction increases MRTF-SRF activity and drives mesenchymal transition

Myh9 modulates actin polymerization, which is known to affect MRTF-SRF transcriptional activity. The SRF co-activator MRTF is bound and inactivated by monomeric G-actin (Fig. 5a). Upon actin polymerization, the pool of monomeric G-actin decreased resulting in the release of MRTF, which activates SRF to induce its target genes[49]. In this respect, we used fluorescent DNaseI, which binds G-actin at femtomolar affinity, as a flow cytometric measure of G-actin levels in isolated nuclei[50]. We found that nuclear monomeric G-actin was diminished in YUMM2.1 cells expressing WT-Bmal1 and dHLH-Bmal1 (Fig. 5b, c). We then sought to determine how the interaction between Bmal1 and Myh9 modulates MRTF-SRF activity in EV, WT-Bmal1, and dHLH-Bmal1 YUMM2.1 cells.

To assess MRTF-SRF activity, we used the MRTF-SRF reporter SRF-RE-luciferase (SRF-RE-LUC containing a CArG box) (Fig. 5a, d). Cytochalasin D (CD) releases MRTF from G-actin, and hence its treatment serves as a strong positive control for the reporter assay (Fig. 5d)[49]. SRF can also be activated by Erk, which stimulates Ets transcription factors to cooperate with SRF at target genes, but Ets competes with MRTF in stimulating SRF activity[51]. On this point, we also inhibited Erk using trametinib, a MEK inhibitor, which enhanced MRTF-driven SRF activity (Fig. 5d). Consistent with the more dramatic transcriptomic changes induced by dHLH-Bmal1 than WT-Bmal1 in cell state shift, SRF-RE-LUC activity was highest in YUMM2.1-dHLH cells (Fig. 5d). Knockdown of both MRTFA and MRTFB reduced SRF-RE-LUC activity (Supplementary Fig. 5a) illustrating their roles as co-factors for SRF activity and validating this assay.

YUMM2.1-WT and YUMM2.1-dHLH as compared with YUMM2.1-EV cells have higher MRTFA protein level, (Supplementary Fig. 5b), which could be a result of increased MRTF-SRF signaling according to the report showing MRTFA is a direct target of MRTF-SRF[49]. Further, the immuno-resistant YUMM1.7 cells[22] (derived from BrafV600E, Pten−/− and Cdkn2a−/− transgenic melanoma[21]) have highest MRTFA and SRF protein and mRNA levels (Supplementary Fig. 5b, c), further illustrating the correlation between MRTFA/SRF levels and resistance to immunotherapy. Further, we found that reducing Myh9 levels with stably expressed shRNAs (Fig. 6a) diminished nuclear G-actin (Supplementary Fig. 6a, b) and enhanced SRF-RE-LUC reporter activity (Fig. 6b). Collectively, these findings are consistent with a model that sequestration of Myh9 by ectopic Bmal1 proteins reduces nuclear G-actin and increases MRTF-SRF activity, which contributes to mesenchymal transition, tumorigenesis and immune evasion.

### Loss of Myh9 function enhances mesenchymal cell state and accelerates tumorigenesis of YUMM2.1

Because shRNA mediated Myh9 knockdown enhances MRTF-SRF activity, we investigated the transcriptomes of YUMM2.1-EV cells that express shRNAs targeting Myh9 (shMyh9) and those of YUMM2.1-EV, YUMM2.1-WT and YUMM2.1-dHLH cells that express control shRNA (shNC). Similar to findings with WT-Bmal1 and dHLH-Bmal1 over-expressed cells (Fig. 3b, c), the EMT gene expression signature was enriched by shMyh9 (Fig. 6c and Supplementary Fig. 6c). We identified 627 overlapped upregulated or downregulated DEGs that were similarly changed by shMyh9 and ectopic expression of WT-Bmal1 or dHLH-Bmal1 as compared with EV (Fig. 6d and Supplementary Fig. 6d). We postulate that these 627 genes illustrate how loss of Myh9 phenocopies ectopic expression of Bmal1 proteins (Fig. 6d). Notably, Sox9 was induced and Sox10 repressed with shMyh9 (Fig. 6e, f), accompanied by corresponding changes in H3K4me3 and H3K27Ac (Supplementary Fig. 6e).

The 413 genes that were induced by WT-Bmal1 or dHLH-Bmal1 and increased by shMyh9 compared to controls (shNC in EV cells; Fig. 6d and Supplementary Data 6) were subject to transcription factor consensus sites analysis, which revealed that AP-1 and SRF motifs were enriched (Fig. 6g). H3K27ac ChIP-seq analysis showed shMyh9 increased H3K27ac signals at 2408 regions and 796 of them overlapped between Bmal1 overexpressed cells (Supplementary Fig. 6f). Further, motif analysis of these 2408 peaks reveals prevalence of AP-1 motif across these regions (Supplementary Fig. 6g), of which 271 are intergenic regions (Supplementary Fig. 6h, i). These changes illustrate the ability of loss of Myh9 to phenocopy the genomic effects of ectopic Bmal1 protein expression (Fig. 3j,k). Congruent with an enrichment of the AP-1 consensus motif, loss of Myh9 increased the expression of selected AP-1 transcription factors either in mRNA or protein levels, such as c-Jun, Junb, Jund, ATF4, ATF5 and ATF6 (Fig. 6h,i). Because Myh9 has been identified as an MRTF-SRF target gene[49] and loss of Myh9 increased MRTF-SRF activity and induced genes enriched in SRF/AP-1 consensus sites, we therefore speculated that Myh9 is involved in a negative feedback loop, in which Myh9 suppresses SRF activity by increasing G-actin levels and thereby inhibiting MRTFA (Fig. 6j).

Given that loss of Myh9 largely phenocopies ectopic expression of Bmal1 proteins, we then sought to determine how decreased Myh9 affects YUMM2.1 tumorigenesis. Notably, Myh9 has been documented to suppress tumorigenesis in mouse models of mammary cancer[41], squamous cell skin[42] and tongue carcinoma[43] and in melanoma[44] without a known common mechanism. We found that knockdown of Myh9 in YUMM2.1 EV (Fig. 7a) increased tumor growth equivalent to WT-Bmal1 (Supplementary Fig. 7a) compared to control shNC YUMM2.1 EV tumors in C57BL/6 mice. Knockdown of Myh9 in WT-Bmal1 cells (Fig. 6a) further modestly increased tumor growth (Supplementary Fig. 7a). Notably knockdown of Myh9 did not enhance growth in either YUMM2.1 EV or YUMM2.1 WT tumors in NSG mice (Fig. 7b and Supplementary Fig. 7b), demonstrating host immune response to tumors was involved in Myh9 mediated tumorigenesis. The modest increased tumor growth by Myh9 knockdown in YUMM2.1 WT cells suggests that ectopic Bmal1 expression did not completely sequester Myh9. Similar but not identical to ectopic Bmal1 expression in YUMM2.1 EV (Fig. 2d), we found that decreased Myh9 increased the

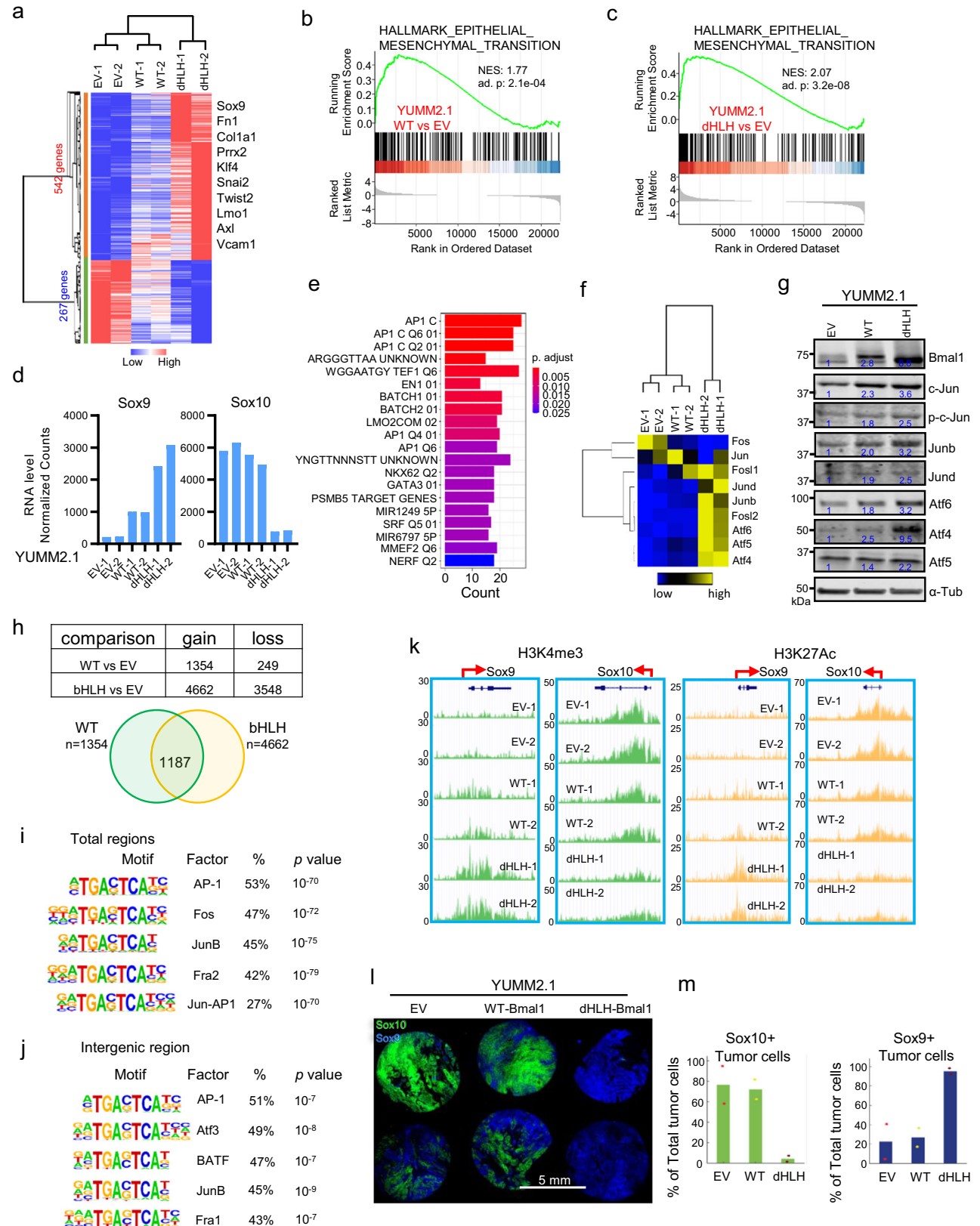

production of VEGF, GM-CSF, IL23, Igfbp6, and KC (Cxcl1) in vitro (Fig. 7c). Immunophenotyping of these tumors revealed increases in M-MDSC, PMN-MDSC, macrophages, and Cd4+ T cells in shMyh9 tumors versus shNC tumors (Fig. 7d), similar to changes seen in tumors driven by dHLH-Bmal1 (Fig. 2d,e). Further, similar to the effect of dHLH-Bmal1 (Fig. 2f), decreased Myh9 expression also rendered YUMM2.1 tumors resistant to anti-PD1 treatment (Fig. 7e). Collectively, our findings indicate that knockdown of Myh9 phenocopies ectopic expression of Bmal1 proteins in YUMM2.1 cells and confers an immune resistant mesenchymal melanoma cell state.

**Fig. 3 | Ectopic Expression of Bmal1 Induces Sox10^high^ YUMM2.1 Cells toward Sox9^high^ More Mesenchymal Cell State.** a–f Analyses of RNA-seq data from YUMM2.1 EV, WT-Bmal1 and dHLH-Bmal1 cells. BR of 2. a Heatmap for all genes that are affected by WT-Bmal1 and dHLH-Bmal1 in the same direction versus EV in YUMM2.1. GSEA showing significantly enriched epithelial mesenchymal transition (EMT) gene set in YUMM2.1 WT-Bmal1 vs EV (**b**) and YUMM2.1 dHLH-Bmal1 vs EV (**c**). Adjust *p*-value by one-sided Fisher's exact test which applied to all GSEA unless noted. **d** Expression of Sox9 and Sox10 mRNA level. **e** Transcription factor motif enrichment among 542 genes that progressively increased from YUMM2.1 EV to WT-Bmal1 and to dHLH-Bmal1 cells shown in Fig. 3a. Adjust *p*-value by one-sided Fisher's exact test which applied to all transcription factor motif enrichment analysis unless noted. **f** Heatmap of AP-1 factors. **g** Immunoblot for Bmal1 and AP-1 factors in YUMM2.1 with EV, WT-Bmal1 and dHLH-Bmal1. Numbers underneath the rows represent relative expression of proteins on different gels but from the same

experiment. RE of 3. h–j H3K27Ac ChIP-seq analysis of YUMM2.1 WT-Bmal1 and YUMM2.1 dHLH-Bmal1 versus YUMM2.1 EV. BR of 2. Gain or loss of H3K27Ac signals in YUMM2.1 cells with WT-Bmal1 and dHLH-Bmal1 versus EV (**h**). Prevalence of bZIP (AP-1) motif in all 1187 regions (**i**) and 131 intergenic regions (>20 kb from TSS) (**j**). Nominal *p*-value by one-sided hypergeometrical test. **k** H3K4me3 and H3K27Ac alterations at Sox9 and Sox10 loci in YUMM2.1 cells with WT-Bmal1 and dHLH-Bmal1 versus EV. BR of 2. l CyCIF for Sox9 (green) and Sox10 (Blue) on tissue microarray of independent tumors from YUMM2.1 EV, WT-Bma1 and dHLH-Bmal1 tumors in C57BL/6 mice. Each circular section was from a different tumor. Source data for CyCIF as Minerva story available at www.cycif.org/data/zhang-2023. **m** Percentage of tumor cells positive for Sox10 and Sox9 respectively in YUMM2.1 EV, WT-Bmal1 or dHLH-Bmal1 tumor tissues determined by CyCIF. Each dot represents one tumor tissue (*n* = 2). BR = biological replicate, RE = replicate experiment. Source data are provided as a Source Data file.

## Discussion

Our study reveals that the role of Bmal1 in tumorigenesis depends on context. First, we observed that YUMM2.1 and B16-F10 Bmal1-null cells have decreased hypoxic Hif1α protein levels, diminished expression of Sox9, and reduced YUMM2.1 tumorigenesis that could be rescued by ectopic Hif1α expression. Secondly, ectopic wild-type and dominant negative Bmal1 proteins sequester Myh9 to induce MRTF-SRF activity, which is associated with a shift of the YUMM2.1 cell line toward an AP-1-associated mesenchymal cell state that resists anti-PD1 therapy. Thirdly, reduction of Myh9 expression induced a mesenchymal state that confers anti-PD1 therapy resistance, phenocopying the effects of ectopic Bmal1 proteins. Collectively, these observations support a model that ectopic expression of WT-Bmal1 or dHLH-Bmal1 sequesters Myh9, modulates SRF activity, and promotes an immune resistant mesenchymal melanoma cell state associated with increased AP-1 enhancer activity (Supplementary Fig. 7c).

The reduction of hypoxic Hif1α protein levels in YUMM2.1 or B16-F10 Bmal1-null cells is consistent with previous studies that propose direct stabilization of Hif1α by Bmal1[27,52]. Importantly, as found with our melanoma syngeneic tumorigenesis studies, Bmal1 loss diminished tumorigenicity in transgenic mouse MLL-AF4-driven leukemia[53], SOS-driven squamous cell skin carcinoma[54], and loss of BMAL1 or CLOCK diminished human glioblastoma tumorigenesis in orthotopic mouse models[55,56]. Paradoxically, BMAL1 positively correlates with antitumor immunity and patient survival in metastatic melanoma[57]. Loss of Bmal1 in the APC^min^ colon cancer model enhanced tumorigenesis[58]. Bmal1 knockdown decreased B16-F10 tumorigenesis in the presence of dexamethasone but not in phosphate buffered saline (PBS)[59]. Increased Bmal1 was associated with diminished B16-F10 tumor growth mediated by melanopsin (Opn4) knockout[60]. BMAL1 also transcriptionally regulates MiTF in human melanoma cells to influence melanin synthesis against UVB irradiation[61]. Given the pleiotropic effects of Bmal1 alterations on tumorigenesis, we speculate that the diverse effects of Bmal1 on tumorigenesis depends on the initial neoplastic cell state and whether Bmal1 modulates tumor initiation versus progression in specific models.

We are intrigued by the neomorphic activity of over-expressed Bmal1 and dominant negative dHLH-Bmal1. These over-expressed proteins shifted YUMM2.1 cells toward a mesenchymal melanoma transcriptional state related to the switch between SOX10 and SOX9 expression observed in human melanomas[33,62]. Notably, we found that human melanomas lacking SOX10 with increased SOX9 expression determined by scRNAseq[35,62] were immune resistant reflecting the anti-PD1 resistant YUMM2.1 Sox10^low^ state observed with ectopic expression of dHLH-Bmal1. Mechanistically, the interaction between Bmal1 or dHLH-Bmal1 and Myh9 was validated through proximity labeling, ligation assays and co-immunoprecipitation. Congruent with the previous finding of nuclear Bmal1-Myh9 binding[46], we found that Bmal1-Myh9 interaction was detected mostly in the nucleus, where nuclear actin is implicated in regulating SRF transcriptional activity[63]. As such,

we speculate from our studies that Bmal1 sequesters nuclear Myh9 to reduce G-actin and induce SRF activity.

SRF has been implicated in human melanoma, particularly with the finding of mutant RAC^P29S^ which increases SRF activity and expression of genes bearing AP-1 or Sox9 consensus binding motifs[64]. The documented role of AP-1 in driving melanoma mesenchymal enhancer activity[33] highlights the importance of MRTF/SRF activation in lineage infidelity and therapy resistance[65]. These reports about human melanomas connect with our unexpected finding of Bmal1-Myh9 interaction driving AP-1 activation and mesenchymal transition[17]. Although our studies primarily focused on murine melanoma and did not address whether clock function is perturbed in human melanomas to induce mesenchymal transition and therapy resistance, a recent study about human prostate cancer demonstrated that human BMAL1 is elevated and necessary for enzulutamide resistance[30]. Future studies are required to address these issues in human melanoma.

Circadian clock and SRF activities have been linked, but the interaction between Bmal1 and Myh9 was not previously recognized. MRTF/SRF was shown to induce Per2, linking SRF to the circadian clock machinery[49]. Conversely, MRTF/SRF level and activity oscillates in a circadian fashion, and AP-1 also has circadian activity in vivo[66,67]. Intriguingly, primary mouse fibroblasts displayed Cry1/Cry2-dependent circadian oscillation of F-actin levels related to time-dependent cell motility and wound healing[68]. Further, loss of Bmal1 decreased F-actin and MRTF/SRF signaling in the C3H10T1/2 adipocytes in vitro[69]. Importantly loss of Bmal1 in vivo dampened MRTF/SRF signaling and stimulated fat beiging. Conversely, in vivo ectopic expression of Bmal1 in murine beige adipocytes increased MRTF/SRF activity associated with impaired beiging[69]. Notably, our observation that overexpressed Bmal1 sequestered Myh9 (myosin IIA), decreased nuclear G-actin, and activated MRTF/SRF signaling are consistent with the observations that myosin II can depolymerize F-actin[70–73]. Hence, we propose that SRF and Bmal1 have interlocking loops that connect the circadian clock, cytoskeletal dynamics, lineage plasticity, tumorigenesis and therapeutic resistance (Supplementary Fig. 7c).

Our study provides a conceptual foundation, connecting Bmal1, Myh9, MRTF/SRF and AP1 signaling, which requires further testing with in vivo genetic studies beyond the documented effect of ectopic Bmal1 on MRTF/SRF and fat beiging[69] and the observation that increased BMAL1 is essential for anti-androgen resistant human prostate cancer[30]. Because Bmal1 activity is affected by oncogenes or hypoxia, we surmise that in vivo cell states and therapeutic resistance could result from intra-tumoral heterogeneity of acidity, oxygenation, and oncogenic functions[5,74,75]. We expect that this conceptual framework will suggest approaches to confront tumor heterogeneity and reduce tumor cell plasticity, perhaps by manipulating the circadian clock for cancer therapy.

We note that our RNA-seq and ChIP-seq studies were replicated twice (*n* = 2) and hence are underpowered. Further, our labeling experiments using TurboID and PLA show proximity of overexpressed

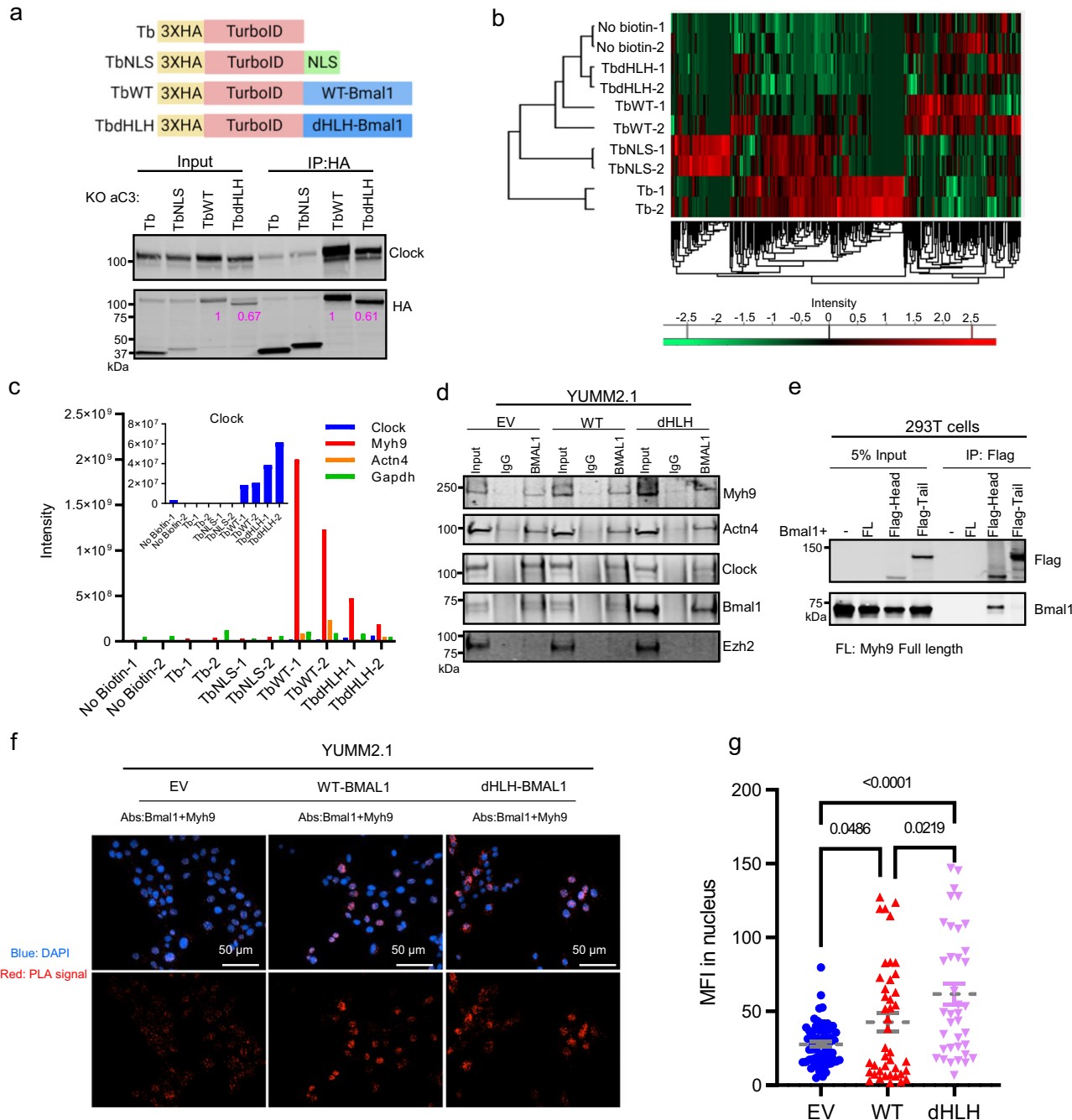

**Fig. 4 | Bmal1 Interacts with Myh9 in Nucleus. a** *Upper panel*: Diagram of TurboID fusion proteins. Tb: 3 Hemagglutinin (HA) tags fused to the 5′ end of TurboID; TbNLS: Nuclear localization signal fused to the 3′ end of Tb; TbWT: WT-Bma1 fused to the 3′ end of Tb; TbdHLH: dHLH-Bma1 fused to the 3′ end of Tb. Created with BioRender.com. *Lower panel*: Immunoblot for HA and Clock with whole cell lysate (Input) and proteins pulled down with HA antibody (IP: HA) from cells: KO aC3-Tb, KO aC3-TbNLS, KO aC3-TbWT and KO aC3-TbdHLH. Numbers underneath the rows represent relative expression. RE of 3. **b** Heatmap of peptide intensities for proteins that were biotinylated, pulled down, and digested from streptavidin beads. Enriched labeled proteins were from cells (shown in Fig. 4a) exposed to biotin and identified by LC-MS/MS analysis. No biotin treated samples were used as negative control for endogenously biotinylated proteins. BR of 2. **c** Peptide intensities for Clock (inset), Myh9, Actn4, and Gapdh are shown from Fig. 4b. **d** Immunoblot of proteins that were co-immunoprecipitated by BMAL1 antibody from nuclear extracts of cross-linked YUMM2.1 EV, YUMM2.1 WT-Bma1 and YUMM2.1 dHLH-Bma1 cells. Normal rabbit IgG was used as antibody control; Clock and Ezh2 were separately used as positive and negative control for immunoprecipitation. RE of 3. **e** Immunoblot of proteins co-immunoprecipitated by Flag antibody from 293 T cells without (−) or with different Myh9 constructs (FL: Full length of Myh9 without Flag tag; Flag-tagged Myh9 Head; Flag-tagged Myh9 Tail) and Bma1 overexpression. RE of 3. **f** Bma1 and Myh9 in situ interaction in YUMM2.1 EV, WT-Bma1 and dHLH-Bma1 detected by Proximity Ligation Assay (PLA) using anti-Bma1 and anti-Myh9 antibodies. Fluorescent micrographs show nuclear staining with DAPI (blue) and PLA signal (red). RE of 2. **g** Mean fluorescence intensity (MFI) of nuclear PLA signals from YUMM2.1 EV (*n* = 66), WT-Bma1 (*n* = 43) and dHLH-Bma1 (*n* = 38) cells. Mean ± SEM. Adjust *p*-value by one-way ANOVA test followed by multiple comparison test. BR = biological replicate, RE = replicate experiment. Source data are provided as a Source Data file.

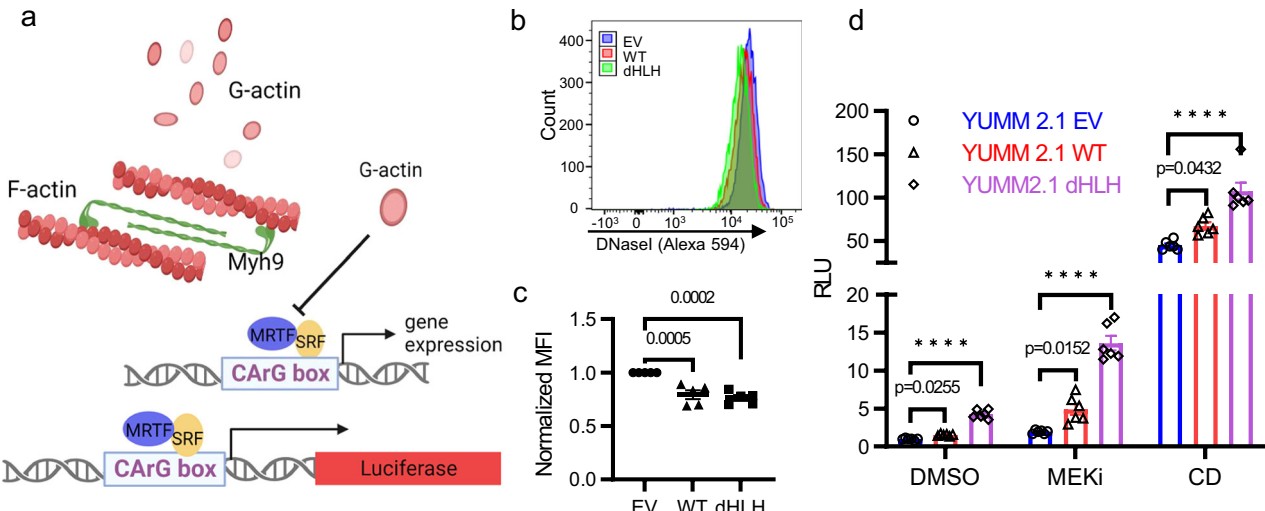

**Fig. 5 | Bmal1 and Myh9 Interaction Increases MRTF-SRF Activity and Drives Cell State Change. a** *Upper panel*: Diagram illustrating the interaction between Myh9, Actin and MRTF-SRF signal pathway. Myh9 (Myosin IIA) formed bipolar filaments binding to F-actin which can depolymerize into monomer G-actin. G-actin can bind to SRF cofactor MRTF to inhibit MRTF-SRF transcriptional activity. *Lower panel*: the diagram of SRF-RE Luciferase reporter to examine the activity of MRTF-SRF pathway. Created with BioRender.com. **b**, **c** Flow cytometry analysis of G-actin stained with Alexa594-conjugated DNaseI in YUMM2.1 EV, YUMM2.1 WT-Bmal1 and YUMM2.1 dHLH-Bmal1 cells. Data (**b**) represents 4 independent experiments which were quantified in (**c**). Data was normalized to EV. Mean ± SEM. Adjust *p*-value by one-way ANOVA test followed by multiple comparison test. **d** Relative luminescence (RLU) from YUMM2.1 EV, WT-Bmal1 and dHLH-Bmal1 cells that were transiently transfected with SRF-RE luciferase and Renilla luciferase plasmids and followed by treatment with 10 nM Trametinib (MEKi) for 20 h or 2 μM Cytochalasin D (CD) for 2 h. DMSO was the 20 h treatment control. Mean ± SEM of 6 BR. RE of 4. Adjust *p*-value by one-way ANOVA test followed by multiple comparisons test. ****$p < 0.0001$. BR biological replicate, RE replicate experiment. Source data are provided as a Source Data file.

Bmal1 proteins with Myh9; however, these data cannot rule out indirect interactions involving other factors, notwithstanding the supportive evidence from immunoprecipitation assays. Altogether, however, biological insights from the sequencing and proteomic data are corroborated by functional luciferase reporter assays and tumorigenesis studies, supporting the connections between Bmal1, Myh9, MRTF/SRF, AP-1, melanoma cell states and immune evasion.

## Methods
### Ethics statement
Our research complies with all relevant ethical regulations. Animal protocols were approved by Institutional Animal Care and Use Committee (IACUC) at Wistar Institute (Protocol number: 201189) and Animal Care and Use Committee (ACUC) at Johns Hopkins University School of Medicine (Protocol number: MO22M452). The maximal tumor volume permitted in our study is 2000 mm³ as calculated from caliper tumor measurements. All procedures were performed in accordance with the protocol and none of the tumor volume in our study exceeded 2000 mm³ at the time point before the last time point when the studies were terminated. Some tumors in Supplementary Fig. 1k exceeded this limit at the last time point, and were immediately euthanized. The age for all mice used in this study is from 7 to 9 weeks old. The ambient temperature in mice holding room was 68–72 °F and humidity was 40–60%.

### Cell culture
YUMM2.1, YUMM1.7 cell lines are from Ashani Weeraratna's lab. B16-F10 Bmal1 knockout clone and control clone are from Amita Sehgal's lab. Parental B16-F10 (Cat#: CRL-6475) and 293 T (Cat#: CRL-3216) cell lines were purchased from ATCC. Human melanoma cell lines are from Meenhard Herlyn's lab. YUMM2.1, YUMM1.7, B16-F10, 293 T and additional cell lines from YUMM2.1, YUMM1.7 and B16-F10 described below were maintained in standard DMEM (4 mM L-glutamine, 25 mM glucose) supplemented with 10% fetal bovine serum and 1X penicillin/ streptomycin in standard humidified 5% CO2, 37 °C tissue culture

incubators. All human melanoma cell lines were maintained in MCDB 153 media supplemented with 20% Leibovitz's L-15 Medium and 2% fetal bovine serum.

### Cell Synchronization
Cells were synchronized by dexamethasone through aspiration of media and replacement with fresh media containing 0.1 μM dexamethasone.

### Luciferase reporter cell lines and monitoring
To generate real-time luciferase reporter cell lines, mouse melanoma cell lines: YUMM2.1 and YUMM2.1 Bmal1 KO clones seeded in 6-well plates were transduced by *Bmal1*::dLUC lentivirus with 8 μg/mL polybrene for 48 h followed by Blasticidin (10 μg/mL) selection for up to 6–8 days. The stably expressed reporter was used to test if these cell lines are clock competent.

To examine the effects of Bmal1 overexpression on luciferase reporter in these cells, YUMM2.1 *Bmal1*::dLUC and YUMM2.1 Bmal1 KO aC3 *Bmal1*::dLUC cells were then seeded in 6-well plate and further transduced with lentivirus to stably overexpress WT-Bmal1 or dHLH-Bmal1. And EV lentivirus was used as a negative control. Stable cell lines expressing WT-Bmal1, dHLH-Bmal1 or EV were sorted for GFP positive by FACS since this lenti-vector is GFP-selectable. The sub cell lines are referred to here as YUMM2.1 EV, YUMM2.1 WT-Bmal1, YUMM2.1 dHLH-Bmal1; KO aC3 EV, KO aC3 WT-Bmal1 and KO aC3 dHLH-Bmal1.

To monitor luciferase signal, reporter cell lines established above were plated in 24-well plates or 35 mm dishes to be confluent at the beginning of analyses. Typically, 50,000 - 70,000 cells per well of 24-well plate or 250,000 - 350,000 cells per 35 mm dish were seeded 1 day prior. At time zero, culture plates or dishes were aspirated, administered fresh Lumicycle media which is DMEM w/o glutamine based (for mouse melanoma cell lines) or RPMI w/o glutamine based (for human melanoma cell lines) supplemented with 5% fetal bovine serum, 4 mM glutamine, 0.1 μM dexamethasone and 0.1 mM beetle potassium luciferin, sealed against desiccation with adhesive optical

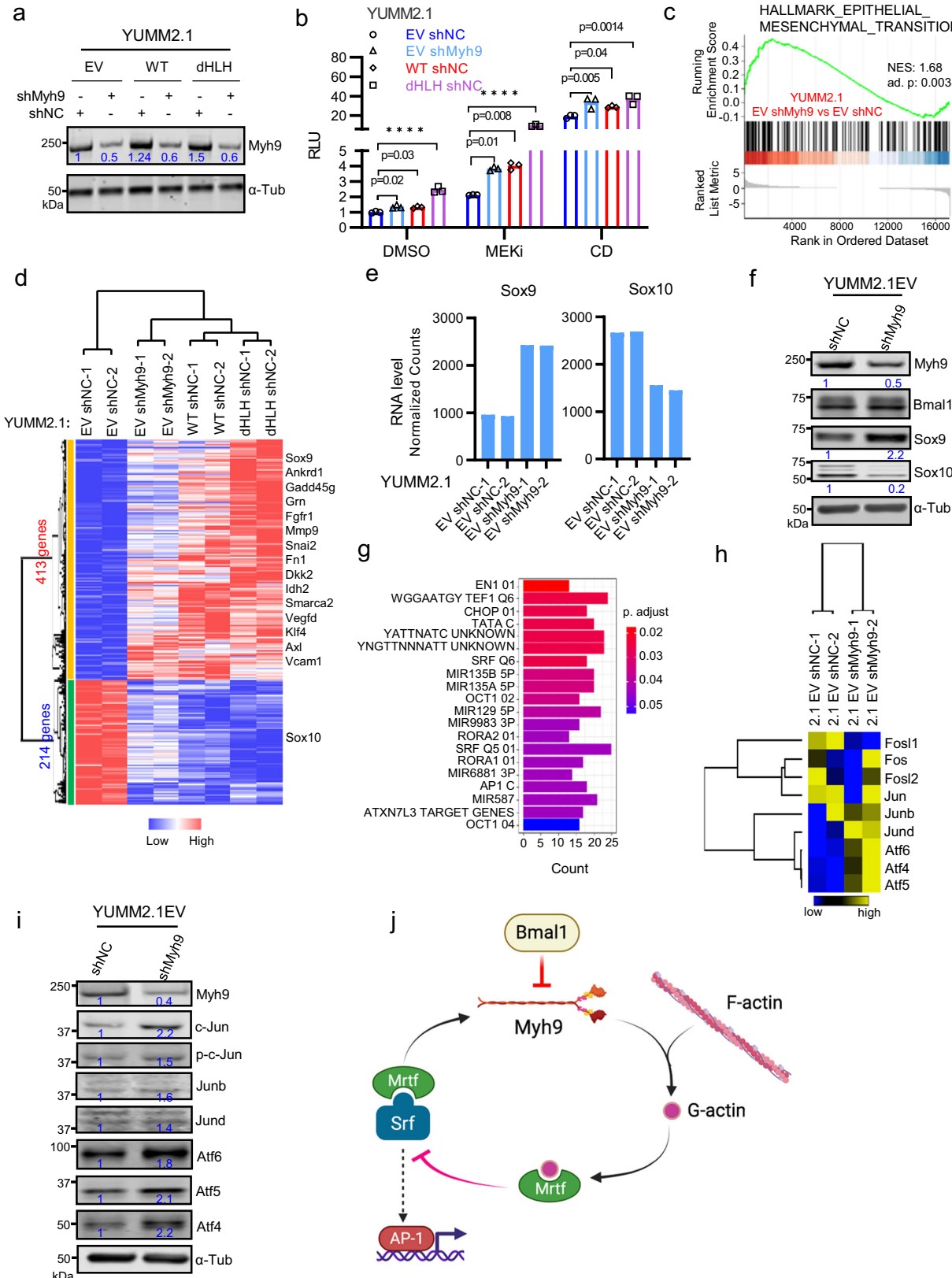

PCR plate film (24-well plate) or vacuum grease (35 mm dishes), and immediately placed in a Lumicycle-96 or Lumicycle-32 luminometer. Luminescence (counts per sec; "relative light units (RLU) per second") was recorded every 10 min for multiple days and exported into Excel with LumiCycle Analysis software. All lumicycle data are generated in atmospheric $CO_2$ conditions and all data presented as detrended.

**CRISPR-editing**

*Bmal1* was knocked out in YUMM2.1 through CRISRP editing using pCRISPR-CG01-sgBmal1-a, pCRISPR-CG01-sgBmal1-b and pCRISPR-CG01-sgBmal1-c from GeneCopoeia (a, b, and c are different guide RNAs against *Bmal1*). pCRISPR-CG01-scramble sgRNA from GeneCo-poeia was used as control. 0.2 million cells were seeded in 6-well plate

**Fig. 6 | Myh9 Knockdown Drives YUMM2.1 Cells to More Mesenchymal Cell State. a** Immunoblot of Myh9 in YUMM2.1 EV, YUMM2.1 WT-Bmal1 and YUMM2.1 dHLH-Bmal1 cells with shNC or shMyh9. RE of 3. **b** Relative luminescence (RLU) from cells EV shNC, EV shMyh9, WT-Bmal1 shNC and dHLH-Bmal1 shNC respectively transiently transfected with SRF-RE luciferase and Renilla luciferase plasmids and treated with 10 nM Trametinib (MEKi) or 2 μM Cytochalasin D (CD) for 20 or 2 h, respectively. DMSO was a control for 20 h treatment. Mean ± SEM of 3 BR. RE of 3. Adjust *p*-value by one-way ANOVA test followed by multiple comparisons test. ****$p < 0.0001$. **c–e** RNA-seq analyses of YUMM2.1 EV with shNC, shMyh9, YUMM2.1 WT-Bmal1 with shNC and YUMM2.1 dHLH-Bmal1 shNC. Samples are from biological duplicates. **c** GSEA showing enrichment of EMT genes with shMyh9 versus shNC in YUMM2.1 EV cells. **d** Heatmap for 627 overlapping genes shown in Supplementary Fig. 5d. **e** Expression Sox9 and Sox10 mRNA level in YUMM2.1 EV cells with shNC and shMyh9. **f** Immunoblot for Myh9, Bmal1, Sox9 and Sox10 in YUMM2.1 EV with shNC and shMyh9. RE of 3. **g** Transcription factor motif enrichment among 413 up-regulated genes shown in Fig. 5d. **h** Heatmap for AP-1 factors from duplicated Quant-seq data of YUMM2.1 EV shNC and shMyh9. **i** Immunoblot of Myh9 and AP-1 factors in YUMM2.1 EV shNC and shMyh9. RE of 3. Numbers underneath the rows represent relative expression of proteins on different gels but from the same experiment. **j** Cartoon illustrating the negative feedback loop in which Myh9 as a target gene of MRTF-SRF, suppresses MRTF-SRF activity through regulating MRTF and G-actin interaction. Created with BioRender.com. BR = biological replicate, RE = replicate experiment. Source data are provided as a Source Data file.

and transiently transfected with 1.5 μg of plasmid using Lipofectamine 2000 according to the manufacture's instruction at the following day. Media was changed with fresh media one day later after transfection and mCherry positive cells were sorted by FACS as single cells into 96-well plates after transfection for 2 ~ 3 days. Single cell clones were then screened by immunoblot for silencing of Bmal1.

## Stable overexpression

To stably overexpress WT-BMAL1, dHLH-BMAL1 and Hif1α-TM in cells YUMM2.1 and YUMM2.1 Bmal1 KO clone aC3, these cells were seeded in 6-well plates and transduced with lentivirus EV (empty vector), WT-Bmal1 and dHLH-Bmal1 with 8 μg/mL polybrene at the following day. Media was changed at the following day after virus infection. When cells were confluent, they were transferred into 10 cm dishes and allowed to expand 2 more days. GFP positive cells were sorted by FACS and overexpression were confirmed by immunoblot. The cell lines are referred to as YUMM2.1 EV, YUMM2.1 WT-Bmal1, YUMM2.1 dHLH-Bmal1, YUMM2.1 Hif1α-TM, YUMM2.1 Bmal1 KO aC3 EV, YUMM2.1 Bmal1 KO aC3 WT-Bmal1, YUMM2.1 Bmal1 KO aC3 dHLH-Bmal1, and YUMM2.1 Bmal1 KO aC3 Hif1α-TM respectively. Also, to stably overexpress TurboID, TurboID-NLS, TurboID-WT-Bmal1, TurboID-dHLH-Bmal1 in YUMM2.1 Bmal1 KO aC3, cells were subject to the similar processes as described above except the lentivirus used for here are TurboID, TurboID-NLS, TurboID-WT-Bmal1 and TurboID-dHLH-Bmal1. And the cell lines are referred to as KO aC3-Tb, KO aC3-TbNLS, KO aC3-TbWT and KO aC3-TbdHLH. B16-F10 EV and B16-F10 Hif1α were made with the same processes using lentivirus EV and Hif1α-TM.

## shRNA knockdown

PLKO.1 vectors expressing 5 different shRNAs against mouse Myh9 or scramble shRNA (shNC) together with packaging plasmids psPAX2 and pMD2.G were transfected into 293 T cells with lipofectamine 2000 to produce 6 different lentivirus which was used to transduce Y21 EV, Y21 WT-Bmal1 and Y21 dHLH-Bmal1 cells separately that were seeded in 6-well plate 1 day prior. After lentivirus transduction for 24 ~ 48 h, media was exchanged with fresh media supplemented with puromycin (2 μg/mL) to start drug selection. Cells were maintained in selection until all non-transduced control cells died off. Knockdown efficiency of these 5 shRNAs was compared with shNC by immunoblot which shows shMyh9#2 has the best knockdown among these 5 shRNAs. The cell lines are referred to as YUMM2.1 EV shNC, YUMM2.1 EV shMyh9#2 (also YUMM2.1 EV shMyh9), YUMM2.1 WT shNC, YUMM2.1 WT shMyh9 and YUMM2.1 dHLH shNC.

## siRNA knockdown

ON-TARGETplus siRNA SMARTPool (referred to as "siRNA") which is a mixture of 4 siRNAs was purchased from Dharmacon and knockdown efficiency was assessed by qPCR and immunoblot. ON-TARGETPlus Non-targeting Control Pool (referred to as "siNT") was also purchased as negative control. 120,000 cells or 12,000 cells were seeded in 6-well

plates or 24-well plates, respectively. The following day, cells were transfected with siRNA or siNT at the concentration of 2.5 nM using Lipofectamine RNAiMAX. Media was changed with fresh media at the second day after transfection. Cells in 6-well plate were maintained and allowed to expand for 2 more days, then were harvested for further immunoblot or RNA extraction experiments. As for the cells in 24-well plate, the cells were treated further to use for Dual-Luciferase Reporter Assay.

## Dual Luciferase assay

20,000 cells were seeded in 24-well plates and were co-transfected with plasmids SRF-RE-dLUC plus Renilla-LUC at the following day using Lipofectamine 2000. 24 h later, aspirated media, and administered fresh media supplemented with or without (for cytochalasin D treatment) 10 nM MEK inhibitor-Trametinib or DMSO used as vehicle control. Cells were maintained for 18 h, then added actin polymerization inhibitor-Cytochalasin D (referred to as "CD") at the concentration of 2 μM to the rest of cells. 2 h later, all cells were rinsed once with PBS and lysed in 100 μL 1X passive lysis buffer. As for the cells need to be treated with siRNA first, please see details described above. Briefly, after cells in 24-well plates were transfected with siRNA 24 h later, they were subjected to the same processes to acquire cell lysate with passive cell lysis buffer. The clear supernatant from the cell lysate was used for luminescence assay with Dual Luciferase Reporter Assay Kit on VIC-TOR Multilable Plate Reader.

## Protein immunoblotting

Cells in 6-well plate were scraped in 1 mL cold PBS after media was aspirated and centrifuged at $400 \times g$ at 4 °C for 5 min in 1.5 mL Eppendorf tubes, cell lysis buffer (Mammalian Protein Extraction Reagent) supplemented with 1X protease inhibitor cocktail, two 1X phosphatase inhibitors were added into cell pallet after PBS was removed. For cells in 3% O2 hypoxia chamber, cell lysis buffer was also supplemented with prolyl hydroxylase inhibitor (200 μM desferrioxamine). Leave cell pallets in cell lysis buffer for at least 20 min on ice. Protein lysates were cleared by centrifugation at $15,000 \times g$ at 4 °C for 15 min. Regarding the immunoblots in Supplementary Fig. 4a, cell cytoplasmic and nuclear fraction was extracted with NE-PER Nuclear and Cytoplasmic Extraction Reagents first according to the manufacture's direction. Then protein yield was quantified with the DC Protein Assay, and same amount of total protein was resolved by SDS-PAGE using Criterion pre-cast Tris-Glycine 4 ~ 20% gradient gels followed by dry transfer to nitrocellulose (NC) membranes with iBlot. NC membranes were blocked with 5% BSA in TBST for 40 ~ 60 min at room temperature. Primary antibodies including anti-Hif1α (1: 1000), anti-Bmal1 (1:1000), anti-Sox9 (1:1000), anti-Sox10 (1:1000), anti-MRTFA (1:1000), anti-MRTFB (1:1000), anti-Myh9 (1:1000), anti-c-MYC (1:10,000), anti-MiTF (1:1000), anti-Clock (1:1000), anti-Actn4 (1:1000), anti-TbP (1:1000), anti-Atf4 (1:1000), anti-Atf5 (1:500), anti-Atf6 (1:1000), anti-c-Jun (1:1000), anti-phospho-c-Jun (1:1000), anti-

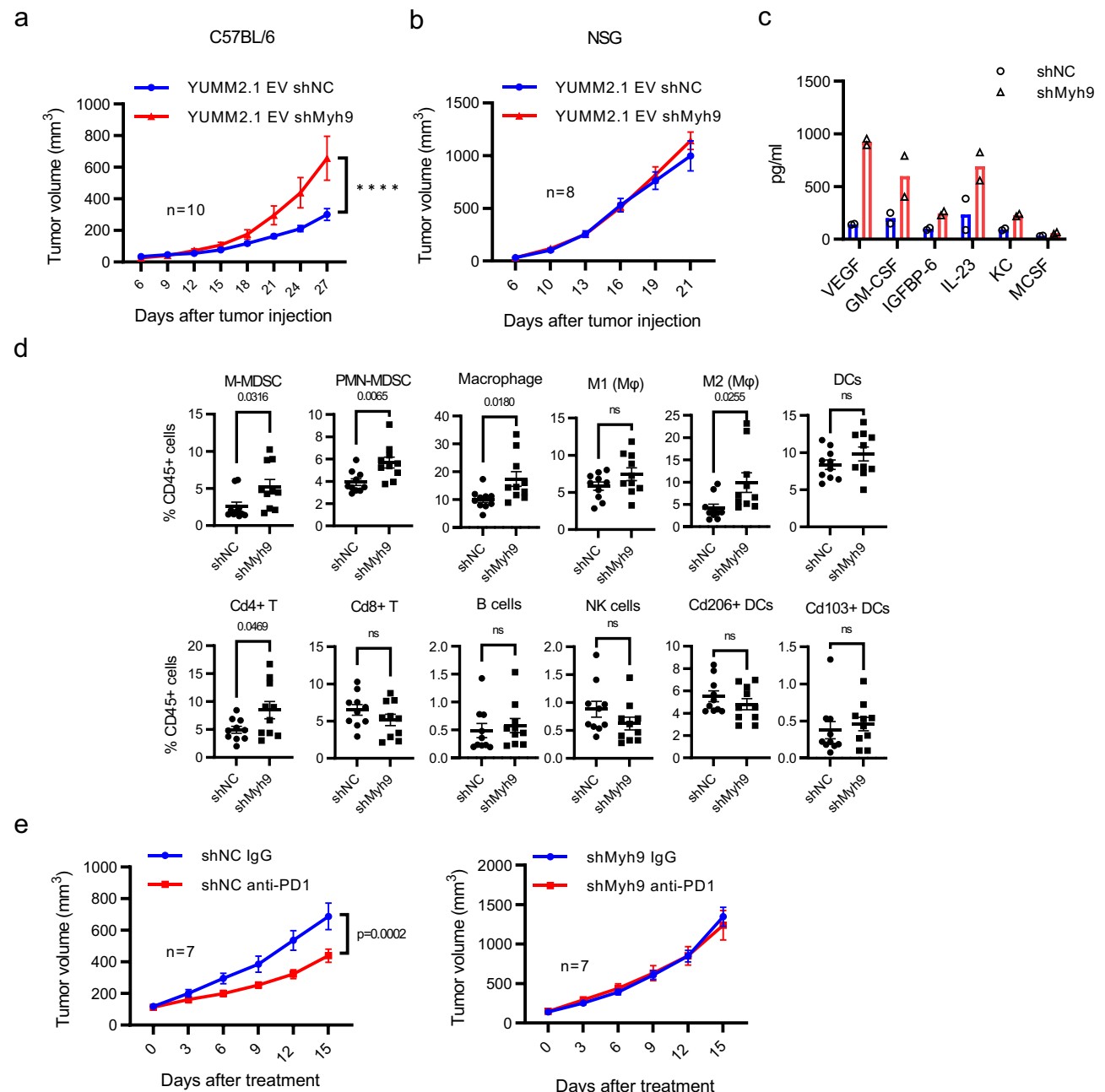

**Fig. 7 | Loss of Myh9 Affects Immune infiltration and Increases Tumorigenesis of YUMM2.1. a** In vivo tumorigenesis of YUMM2.1 EV with shNC or shMyh9 in male C57BL/6 mice (*n* = 10 each group). Mean ± SEM. ****\*p*-value < 0.0001 by Two-way ANOVA test. RE of 2. **b** In vivo tumorigenesis of YUMM2.1 EV with shNC or shMyh9 in male NSG mice (*n* = 8 each group). Mean ± SEM. RE of 2. **c** The level of cytokines secreted from YUMM2.1 EV cells with shNC or shMyh9 as determined by immunoassay. BR of 2. Each dot represents one biologically independent sample. **d** Flow cytometric immunophenotyping of tumors formed with YUMM2.1 EV with shNC or shMyh9 cells in C57BL/6 mice (Fig. 7a). Mean ± SEM. Two-tailed *p*-value by unpaired *t*-test. **e** Response of YUMM2.1 EV shNC and shMyh9 tumors in C57BL/6 mice to control IgG and anti-PD1 treatment given IP every 3 days (*n* = 7 each group). Mean ± SEM. *p* value by Two-way ANOVA test. BR = biological replicate, RE = replicate experiment. Source data are provided as a Source Data file.

Junb (1:1000), anti-Jund (1:1000), anti-α-tubulin (1:10,000) were diluted in blocking buffer and incubated overnight. Secondary antibodies including Alexa Fluor 790 goat anti-mouse IgG (H + L) and Alexa Fluor 680 goat anti-rabbit IgG (H + L) were incubated for 1 h at room temperature. Immunoblots were imaged with Odyssey CLx infrared imaging system (LI-COR) and uniformly contrasted. The quantification of immunoblot was calculated with software image studio version 5.2.5. Please see all uncropped immunoblots scans in the Source Data file and Supplementary Fig. 8.

**Protein immunoprecipitation**
1.5 million cells were seeded in 10 cm dishes and allowed to expand in 10 mL normal media in 37 °C, 5% CO2 incubators. Two days later, cells were crosslinked with 1% formaldehyde for 10 min, neutralized with glycine for 5 min and washed twice with cold PBS, then scrapped into 1 mL cold PBS with protease inhibitor cocktail (PIC, Cell signaling #7012) and transferred into 1.5 mL tubes. To extract nuclei, cells were centrifuged at 300 × *g*, then resuspended in 1.3 mL 1X buffer A (Cell signaling, 4X, #7006) supplemented with 1X PIC and 0.5 mM DTT and

incubated on ice for 10 min with mixing by inverting tubes every 3 min. Nuclei were centrifuged at $2000 \times g$ for 5 min at 4 °C and washed by resuspending in 1.3 mL ice-cold 1X buffer B (Cell signaling, 4X, #7007) with 0.5 mM DTT. After washing, nuclei were centrifuged again at $2000 \times g$ for 5 min at 4 °C and resuspended in 0.5 mL high salt nuclear extraction buffer (10 mM HEPES pH 8.0, 25% glycerol, 1.5 mM MgCl2, 0.1 mM EDTA, 0.2 M NaCl, 1X PIC, 0.5 mM DTT), and incubated on ice for 20 min. Then nuclei extract was sonicated 12 pulses (5 pulses per time) with Branson Sonifier 450 at setting Duty cycle: 50% constant, Output Control: 4. Samples were incubated on ice during and between pulses. Nuclei lysates were clarified by centrifugation at $16,000 \times g$ for 15 min at 4 °C. Protein concentration was quantified with DC Protein Assay. 40 μg nuclei lysate from each sample was diluted with high salt nuclear extraction into 40 μL and used as "Input" control while 800 μg lysate was also diluted with high salt nuclear extraction buffer into 800 μL and pre-cleared with 40 μL Dynabeads ™ Protein A (Invitrogen, 10001D) at room temperature for 40 min. Then this 800 μL lysate was split into 2, followed by addition of IgG or BMAL1 antibody separately. The immunoprecipitation samples were incubated overnight at 4 °C. Then 20 μL magnetic protein A beads were added into each of them and incubated at room temperature for 1 h. Then magnetic protein A beads bound with immuno-complex were separated by placing the tubes in a Magnetic Separation Rack and washed 3 times with high salt nuclear extraction buffer. Proteins were eluted by boiling the beads at 95 °C for 7 min in 50 μL 1X protein loading buffer. All proteins from immunoprecipitated samples and "Input" controls were resolved by SDSD-PAGE and immunoblotted with antibodies as described above in the section of protein immunoblotting.

## Chromatin Immunoprecipitation (ChIP) sequencing and data analysis

ChIP experiments were performed with SimpleChIP Enzymatic Chromatin IP Kit (Magnetic Beads, Cell Signaling #9003) according to the manufacturer's instructions. For sample preparation, cells seeded in 15 cm dishes were processed and nuclei were extracted as described above in the section of Protein Immunoprecipitation. After nuclei pallet was resuspended in 1X buffer B with 0.5 mM DTT, Micrococcal Nuclease was added in and incubated at 37 °C for 20 min with frequent mixing to digest DNA to length of approximately 150 - 900 bp, then digestion was terminated by addition of 0.5 M EDTA and placing tubes on ice for 1 - 2 min. Digested nuclei pallet was resuspended in 1X ChIP buffer with 1X PIC after centrifugation at $16,000 \times g$ for 1 min at 4 °C and supernatant removal. Then nuclei lysates were sonicated 20 pulses with Branson Sonifier 450 at the setting Duty cycle: 50% constant, Output Control: 4. Samples were incubated on ice during and between pulses. Then lysates were cleared by centrifugation at $9400 \times g$ for 10 min at 4 °C, and supernatant (cross-linked chromatin preparation) was transferred into a new tube. 50 μL supernatant was transferred into another tube and used to examine the quality of Chromatin Digestion and DNA concentration after further RNase A, proteinase K digestion, DNA purification and 1% agarose gel resolution. 7 μg of digested, cross-linked chromatin was used for each immunoprecipitation (IP). Antibodies against H3K4me3 (10 μL), H3K27me3 (10 μL), H3K27Ac (10 μL), H3 (positive control) and Rabbit IgG (negative control) were added into samples separately and incubated at 4 °C overnight with rotation. After incubation with 30 μL ChIP-Grade Protein G Magnetic beads at 4 °C for 2 h, the tubes were placed in a Magnetic Separation Rack to separate the magnetic beads and remove supernatant. Then the beads were washed 3 time with low salt buffer and once with high salt buffer at 4 °C with rotation. Chromatin was eluted from the antibody/protein G beads by incubating at 65 °C for 30 min with gentle vortex. To reverse cross-links, 6 μL of 5 M NaCl and 2 μL of Proteinase K were added into the cleared supernatant which contains eluted chromatin and incubated at 65 °C for 2 h. And DNA from all samples was purified using spin columns and stored at −20 °C for qPCR and sequencing.

To sequence the DNA samples purified above, 2.5 ng of ChIP DNA was used to prepare libraries for Next Generation Sequencing using the NEBNext Ultra II DNA kit (New England Biolabs, Ipswich, MA), according to manufacturer's directions. Overall library size was determined using the Agilent Bioanalyzer 2100 and the High Sensitivity DNA assay and libraries were quantitated using the Qubit 2.0 Fluorometer (Thermofisher, Waltham, MA). Libraries were pooled and High-Output, Single read, 75 base pair Next Generation Sequencing was done on a NextSeq 500 (Illumina, San Diego, CA).

After sequencing, ChIP-seq data was aligned using bowtie[76] against mm10 version of the mouse genome and HOMER[77] was used to call significant peaks using "-style histone" option and for generation of bigwig signal files normalized to number of reads per bp per 10 M sequenced reads. Normalized signals for regions were derived from bigwig files using bigWigAverageOverBed tool from UCSC toolbox with mean0 option. Association of expression and histone modification signal changes were performed using 3 kb region around gene TSS for H3K4me3, H3K27Ac and gene body for H3K27me3. Also, H3K27Ac signals from regions >20 kb around gene TSS were analyzed. In addition, motif enrichment analysis within H3K27ac peaks was done using HOMER[77] within its known Motif dataset with -size 1000 parameter option. Top 5 significant results sorted by motif prevalence were reported.

## Proximal ligation Assay (PLA)

All PLA experiments were performed with Duolink In Situ Red Starter Kit Mouse/Rabbit (Millipore Sigma, DUO 92101). In brief, 3000 - 4000 cells were seeded in each well on "PTFE" Printed Slides (10 well, 6 mm well diameter Cat. #63424-06) and allowed to adhere and grow in normal media in 5% CO2 incubators for 1 day. Cells were then rinsed once with PBS and fixed with fresh 4% paraformaldehyde in PBS for 15 min at room temperature. Cells were then rinsed twice with PBS before permeabilizing for 7 min with 0.1% Triton X-100 in PBS. Cells were then washed twice with PBS and blocked with blocking buffer overnight at 4 °C. Primary antibodies anti-BMAL1 (1:200, Santa Cruz, Cat. #sc365645), anti-CLOCK (1:200, Cell Signaling, Cat. #5157), anti-Myh9 (1:200, Proteintech, Cat. #11128-1-AP), anti-Filamin A (1:100, Abcam, Cat. #ab76289), anti-Ki67 (1:500, Cell Signaling, Cat. #9129), anti-HA (12CA5, 2 μg/mL, Millipore Sigma, Cat. #11583816001) were diluted in the Duolink Antibody Diluent in combinations: anti-BMAL1+ anti-CLOCK, anti-BMAL1+ anti-MYH9, anti-HA+ anti-MYH9, anti-HA+ anti-FLNA, anti-HA+ anti-CLOCK, anti-HA+ anti-Ki67, anti-BMAL1+ anti-FLNA, only anti-MYH9 and added to cells, incubated at 4 °C overnight. Then slides were washed 3 × 5 min with 1X Wash Buffer A at room temperature. Then the PLUS and MINUS PLA probes diluted at 1:5 in the Duolink Antibody Diluent were added to slides and incubated in a 37 °C pre-heated humidity chamber. 1 h later, slides were washed 3 × 5 min with 1X Wash Buffer A, and ligase in 1X ligation buffer was added to the slides and incubated in the 37 °C pre-heated humidity chamber for 30 min. After slides were washed 2 × 5 min with 1X Wash Buffer A, polymerase at a 1:80 dilution in 1X Amplification Buffer was added to the slides and incubated in the 37 °C pre-heated humidity chamber for 100 min. From this step, slides were kept in dark to protect from light. Then slides were washed 2 × 10 min with 1X Wash Buffer B followed by one wash with 0.01X Wash Buffer B for 1 min. Slides were mounted with a coverslip using a minimal volume of Duolink PLA Mounting Medium with DAPI. Images were acquired on Nikon 80i Upright Microscope using 40X objective (Fig. 4f, Supplementary Fig. 4b, c) or on ECHO Revolve Microscope using 20X objective (Supplementary Fig. 4d). Mean fluorescence intensity of PLA signals in nucleus was analyzed by Image J.

## Proteomics

To prepare biotinylated samples, 1.5 million cells of each cell line Y21 Tb, Y21TbNLS, Y21 TbWT and Y21 TbdHLH were seeded and grown in

10 cm plates in 37 °C, 5% CO2 incubators. Two days later, biotin was added to the cells at the concentration of 500 μM and incubated for 2 h to label the samples. Labeling was stopped by placing cells on ice and washing 5 times with ice-cold PBS. Cells were scraped into 1 mL ice-cold PBS, transferred into 1.5 mL tubes and palleted by centrifugation at $300 \times g$ for 5 min at 4 °C. Supernatant was removed, and pallets were resuspended and lysed in 400 μL 1X RIPA lysis buffer (Cell Signaling, #9806) supplemented with 1X protease inhibitor cocktail, two 1X phosphatase inhibitors for 10 min followed by 20 pulses (5 pulses per time) sonication with Branson Sonifier 450 at setting Duty cycle: 20% constant, Output Control: 3. Lysates were clarified by centrifugation at $15,000 \times g$ at 4 °C for 15 min. To pull down biotinylated proteins, 15 μL Pierce Streptavidin Magnetic Beads (Thermo Fisher, #88816) were washed twice with 1X RIPA buffer, incubated with 400 μL clarified lysates containing 400 μg protein from each sample with rotation for 1 h at room temperature, then moved to 4 °C and incubated overnight with rotation. After that, the beads were subsequently washed twice with 1 mL of RIPA lysis buffer (SDS final Conc. increased to 0.5%), once with 1 mL of 1 M KCl, once with 1 mL of 0.1 M $Na_2CO_3$, once with 1 mL of 2 M urea in 10 mM Tris-HCl (pH8.0), and twice with 1 mL RIPA lysis buffer. Biotinylated proteins were then eluted from the beads by boiling the beads (95 °C for 5 min) in 80 μL of 1X protein loading buffer supplemented with 5% 2-mercaptoethanol and 2 mM biotin. Then the elutes were transferred into new 1.5 mL tube and kept in −80 °C. At last, the beads were washed 3 times with 50 mM ammonium bicarbonate to remove the SDS, then the dry beads were also kept in −80 °C for further processing and liquid chromatography tandem mass spectrometry (LC-MS/MS) analysis.

LC-MS/MS analysis was performed on a Q Exactive Plus mass spectrometer (ThermoFisher Scientific) coupled with a Nano-ACQUITY UPLC system (Waters). Biotinylated proteins tightly bound to streptavidin beads were digested on-bead with trypsin in 8 M urea, 20 mM glycine, 100 mM Tris-Cl, pH 8.0, and injected onto a UPLC Symmetry trap column (180 μm i.d. x 2 cm packed with 5 μm C18 resin; Waters). Tryptic peptides were separated by reversed phase HPLC on a BEH C18 nanocapillary analytical column (75 μm i.d. x 25 cm, 1.7 μm particle size; Waters) using a 95 min gradient formed by solvent A (0.1% formic acid in water) and solvent B (0.1% formic acid in acetonitrile). A 30-min blank gradient was run between sample injections to minimize carryover. Eluted peptides were analyzed by the mass spectrometer set to repetitively scan m/z from 400 to 2000 in positive ion mode. The full MS scan was collected at 70,000 resolution followed by data-dependent MS/MS scans at 17,500 resolution on the 20 most abundant ions exceeding a minimum threshold of 20,000. Peptide match was set as preferred, exclude isotopes option and charge-state screening were enabled to reject singly and unassigned charged ions.

Peptide sequences were identified using MaxQuant 1.6.3.3[78]. MS/MS spectra were searched against the UniProt mouse protein database (Oct 20019), the recombinant protein sequences and a common contaminant database using full tryptic specificity with up to two missed cleavages, static carbamidomethylation of Cys, and variable oxidation of Met, deamidation of Asn, protein N-terminus carbamylation, and protein N-terminal acetylation. Consensus identification lists were generated with false discovery rates of 1% at protein, and peptide levels.

### RNA collection
All RNA samples were extracted with TRIzol Reagent (Invitrogen, Cat. #15596018) according to the manufacture's instruction and cleaned by DNA-free DNAase Treatment and Removal Reagents (Ambion, #AM1906) to get rid of DNA contamination before submission for RNA sequencing.

### Quantification of nuclear G-actin
Nuclei were isolated then resuspended in PBS, fixed in 4% paraformaldehyde for 15 min at R.T and permeabilized with 0.2% Triton X-100 in PBS for 10 min at R.T. Then, stain these nuclei with Alexa594-conjugated DNAseI (0.3 μM) for 1.5 h at 37 °C under continuous shaking. After resuspended in PBS containing 1 mM EDTA and 1 mM $MgCl_2$, data were acquired in BD FACSymphony and analyzed with FlowJo software (version 10.8.1). Mean of fluorescence intensity (MFI) was calculated and normalized to the control.

### Quant 3′ mRNA-sequencing and data processing
3′ mRNA-seq libraries were generated from 100 ng of DNAseI treated total RNA using the QuantSeq FWD Library Preparation kit (Lexogen, Vienna, Austria), according to manufacturer's directions. Overall library size was determined using the Agilent Tapestation and the D5000 Screentape (Agilent, Santa Clara, CA). Libraries were quantitated using real-time PCR (Kapa Biosystems, Wilmington, MA). Libraries were pooled and High-Output, Single read, 75 base pair Next Generation Sequencing was done on a NextSeq 500 (Illumina, San Diego, CA).

RNA-Seq data was aligned using bowtie2[79] against mm10 version of the mouse genome and RSEM v1.2.12 software was used to estimate raw read counts and FPKM values using Ensemble transcriptome information. DESeq2[80] was used to estimate significance of differential expression between groups of samples. Overall gene expression changes were considered significant if passed FDR < 5% unless stated otherwise.

In addition, gene set enrichment analysis (GSEA) was performed using the R packages msigbdr, clusterprofiler and enrichplot. Briefly, gene expression data from Quant-seq were averaged across different samples with log2 fold-change calculated pairwise. Gene sets were retrieved from the MSigBD (https://www.gsea-msigdb.org/gsea/msigdb/index.jsp) database using the R package msigdbr. Here, the hallmark gene sets (H) of MsigDB was used. Next, clusterprofiler was used to identify enriched gene sets using the pairwise log2 fold-change of expression calculated previously. Enrichment of specific gene signatures, such as hypoxia, epithelial-mesenchymal-transition were visualized using enrichplot. Enrichment of all gene sets within a pairwise comparison was represented in a bubble plot which was generated using the R package ggplot2.

Consensus transcription factor motif enrichment analysis of the RNA-seq data was performed with DEGs as input. Gene sets based on predicted transcription factor binding motifs, which are the regulatory target gene sets (C3) of MSigDB, were retrieved using the R package msigdbr. Enrichment of transcription factor binding motifs among the upregulated or downregulated genes was determined using the "enricher" function of the R package clusterProfiler and was plotted as bar charts using the R package ggplot2.

### In vivo tumor growth and anti-PD1 treatment
To form tumors subcutaneously, 1 million cells suspended in 100 μL PBS and Matrigel (2:1) per mouse were injected into the flanks for C57BL/6 or NSG mice. Tumor volume was measured with caliper from day 6 after injection every 2, 3 or 4 days. For the experiments with anti-PD1 treatment, at the day 17 after injection, size-paired tumors were treated with intraperitoneally (i.p.) injected 300 μg anti-PD1 (Invivo-Mab, RMP1-14) or isotype control (InVivoMab, 2A3) every 3 days for 4 times accompanying tumor measurements (Supplementary Fig. 2f). In parallel, to avoid the tumor size effects on anti-PD1 response, antibody treatment was also started when tumor volume reached about 100 mm³, and 6 doses of 300 μg of anti-PD1 or isotype control were i.p. injected every 3 days into mice with size-paired tumors (Figs. 2f and 6e).

### Tissue-based cyclic immunofluorescence (t-CyCIF)
t-CyCIF was performed as described in (Lin et al., 2018; Burger et al., 2021) and at protocols.io (dx.doi.org/10.17504/protocols.io.bjiukkew). FFPE slides were baked at 60 °C for 30 min using BOND RX Automated

IHC/ISH Stainer and dewaxed using Bond Dewax solution at 72 °C. Antigen retrieval was performed using Epitope Retrieval 1 (LeicaTM) solution at 100 °C for 20 min. For multiplexed imaging, each slide underwent multiple cycles of antibody incubation, imaging, and fluorophore inactivation. Antibodies along with Hoechst 33342 (0.25 μg/mL; LI-COR Biosciences) for DNA staining were diluted in Odyssey Intercept Buffer, and the slides were incubated in primary antibodies (1:100 dilution) overnight at 4 °C in the dark. Glass coverslips were wet mounted before imaging using 250 μL of 70% glycerol in 1X PBS. Images were acquired using the CyteFinder II HT Instrument, an automated slide scanning fluorescence microscope (RareCyte Inc. Seattle WA) with a 20X /0.75 NA objective. After image acquisition, slides were soaked in 42 °C PBS to facilitate coverslip removal. After slides were decoverslipped, they were incubated in a solution of 4.5% H2O2 and 20 mM NaOH in PBS under an LED light source for 45 min X 2 for fluorophore inactivation.

### image processing and data analysis for t-CyCIF

To mitigate image acquisition artifacts, illumination correction was first performed on image tiles using the BaSiC algorithm. Then ASHLAR (PMID: 35972352), Alignment by Simultaneous Harmonization of Layer/Adjacency Registration, was used to stitch these image tiles and register each IF cycle together into a single OME-TIFF file consisting of an image pyramid with multiple resolutions. Full codes are available at: DOI: 10.5281/zenodo.10182504. Post image registration, each image was imported into OMERO, a highly-multiplexed visualization software, for detailed inspection of both cycle registration and antibody staining. More information on the software can be found here: https://www.openmicroscopy.org/omero/. Omero was used to identify and export regions of interest. Regarding cell segmentation, ilastik, a machine learning based (bio) image analysis tool, was used to generate nuclear and cytoplasmic probability masks from OME-TIFF files (PMID: 31570887). To increase the processing speed, 250 × 250 pixel regions from the original OME-TIFF were randomly selected and manually annotated to train a Random Forest based classifier. This model was then applied to the entire image. The obtained masks describe the probability that each pixel belongs to nuclear, cytoplasmic, or background areas. Binary masks for nuclear and cytoplasmic areas were generated using a MATLAB (version 2019a) script for cell segmentation that thresholds on these probability masks and then performs water shedding.

The pre-processing steps of the data analysis that include data aggregation, filtering, and normalization were performed as in Gaglia et al. 2022 (PMID: 35292783). First, for data aggregation, the output file (.mat) from the processing workflow contains single cell data for each IF channel imaged, morphological features, and any metadata. The data matrices from each.mat files are concatenated into a single matrix for measured metrics (median/mean and nuclear/cytoplasmic) into a single structure ('AggrResults') and morphological data (area, solidity, and centroid coordinates) are concatenated into a single structure ('MorpResults'). MorpResults also contains an ID vector that keeps track of which tissue each cell belongs to. Then, as for data filtering, to exclude segmentation errors and cells lost throughout imaging from downstream analysis, the single cell data is filtered based on both morphological and DAPI-based criteria, which are calculated for each IF cycle. Desired ranges were set for parameters such as nuclear area and cytoplasmic area, and minimum cutoffs for other features including: nuclear solidity, absolute DAPI intensity, and the ratio between nuclear and cytoplasmic DAPI measurements. Filter information is allocated to a logical (0–1) structure 'Filter', and used to select cells for further analysis by indexing. Threshold selection is dataset dependent and performed based on data inspection. Last, to normalize the data, for every channel, the probability density function of the log2-transformed data is altered in two ways: (1) the center of the

distribution is shifted to 0 and (2) the distribution width is rescaled. This normalizes each channel in such a way to facilitate cross-channel comparisons. This standard normalization is performed using a two-component Gaussian mixture model, each model capturing the negative and the positive cell population. If this model fails to approximate the channel distribution, two other strategies are attempted: (1) a three-component model is used assuming the components with the two highest means are the negative and positive distribution (this discards the lowest component) or (2) the user selects a percentage x of assumed positive cells and a single Gaussian distribution fit is performed on the remainder of the data to capture the negative distribution. The single Gaussian fit is then used as the lower component in a two-component model to estimate positive population distribution.

### Tumor dissociation

Tumor tissues from Figs. 2b, 7a were dissociated using Mouse Tumor Dissociation Kit (Miltenyi Biotec, #130-096-730) according to the manufacturer's instructions. In short, after tumors were dissected and rinsed with ice-cold PBS, 0.2 - 0.4 g tumor tissues without fat, fibrous or necrotic areas from each sample were cut into small pieces of 2 - 4 mm and transferred into the gentleMACS C tubes containing the enzyme mix which contains 100 μL Enzyme D, 10 μL enzyme R, and 12.5 μL Enzyme A and diluted in 2.35 mL DMEM. Put the tubes into the gentleMACS Octo Dissociator and ran program 37C_m_TDK_1 with Heaters. 22 min later, detached the tubes from the dissociator, resuspended the samples and applied the cell suspension to a cell strainer (70 μm) placed on a 50 mL tube. Then the cell strainers were washed with 15 mL DMEM, and cells were centrifuged at 300 × g for 10 min at 4 °C. 10X Red Blood Cell lysis buffer (eBioscience, #00430054) was diluted to 1X and used to remove erythrocytes for flow cytometry analysis.

### Immunophenotyping of tumor tissues

After tumor tissues were dissociated into single cell and erythrocytes cleaned as described above, about 8 - 16 million cells from each sample were resuspended in 200 μL staining buffer and separated into 2 FACS tubes for myeloid cells and lymphocytes antibodies labeling respectively. 2 μL rat anti-mouse CD16/CD32 (1:50, BD #553141) antibody was added into each sample and incubated with cells on ice for at least 5 min. Meanwhile, antibody panel for myeloid cells: TCRβ FITC (1:100, Biolegend #109205, Clone #H57-597), CD19 FITC (1:100, Biolegend #115505, Clone #6D5), CD11b BV785 (1:100, Biolegend #101243, Clone #M1/70), Ly6G APC Cy7 (1:100, Biolegend #127623, Clone #1A8), Ly6C PerCPCy5.5 (1:100, BD #560525, Clone #AL-21), F4/80 PE (1:100, eBioscience #12-4801-80, Clone #BM8), CD11c APC (1:100, BD #561119, Clone #HL3), MHCII (I-A/I-E) BV605 (1:100, Biolegend #107639, Clone #M5/114.15.2), CD103 BV421 (1:100, Biolegend #121421, Clone #2E7), CD206 PE Cy7 (1:100, Biolegend #141719, Clone #C068C2) and antibody panel for lymphocytes which includes Ly6G-FITC (1:100, Biolegend #127605, Clone #1A8), CD11c FITC (1:100, BD #557400, Clone #HL3), CD3 BV785 (1:50, Biolegend #100355, Clone #145-2C11), NK1.1 PE (1:100, BD #557391, Clone #PK136), B220 PerCPCy5.5 (1:100, BD #561101, Clone #RA3-6B2), CD8 APC Cy7 (1:100, BD #561967, Clone #53-6.7), CD4 BV605 (1:100, Biolegend #100451, Clone #GK1.5) were used to make master mix separately, In addition, antibody CD45 Alexa Fluor 700 (1:100, Biolegend #103128, Clone #30-F11) and Live/Dead dye Aqua (Thermo Fisher #L34965) were added into both master mixes. The master mix of antibodies and Aqua was protected from light and added into corresponding tubes to incubate with cells for 30 min on ice. Cells were centrifuged at 500 × g for 10 min at 4 °C and washed once with 1 mL flow staining buffer to get rid of residual antibodies. Then, cells were resuspended in 0.5 mL flow cytometry buffer to run on FACS Symphony A3, and acquired data were analyzed with FlowJo software.

## Quantification and statistical analysis

Statistical analyses in this study were performed using GraphPad Prism 9 software. For mouse experiments, the exact number of mice ($n$), tumor growth measurements over time (mean ± SEM) and statistical significance are reported in the figures and figure legends. Tumor growth comparisons were analyzed by Two-way ANOVA with Tukey's multiple comparisons test. For comparing tumor weight, one-way ANOVA followed by Tukey's multiple comparisons test was used. For immunophenotyping data analysis, we performed unpaired $t$-test (only two conditions) or one-way ANOVA (more than 2 conditions) followed by Tukey's multiple comparisons. The statistical analyses for SRF-RE luciferase reporter assay were performed with one-way ANOVA test followed by multiple comparison test. Real-time luciferase reporter assay presented as the mean of biological replicates with standard error of mean (SEM) calculated by Microsoft Excel.

More information about reagents used in this study can be found in Supplementary Table 1 in Supplementary Information.

## Reporting summary

Further information on research design is available in the Nature Portfolio Reporting Summary linked to this article.

## Data availability

All RNA-seq and ChIP-seq data analyzed in our study were deposited into GEO database with accession number GSE202289. The mass spectrometry proteomics data have been deposited into the ProteomeXchange (https://www.proteomexchange.org/) repository with the accession number PXD037077. Public Single cell RNA-seq data from Single Cell Portal (Study: Melanoma immunotherapy resistance) were analyzed (https://singlecell.broadinstitute.org/single_cell/study/SCP109/melanoma-immunotherapy-resistance). The remaining data are available in this Article, Supplementary Information and Supplementary Data File. Source data are provided with this paper.

## Code availability

Image data and code associated with t-CyCIF are available at Zenodo (https://doi.org/10.5281/zenodo.10182504).

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

## Acknowledgements

We thank Sonali Majumdar and Sandy Widura (Genomics core at Wistar Institute) for their help with library prep for RNA sequencing, ChIP sequencing and scRNA sequencing, Jayamanna Wickramasinghe and Bhanu Chandra Karisetty (Bioinformatics core at Wistar Institute) for Heatmap plot of Supplementary Fig. 3d and RNA-seq analysis related to Fig. 1a and supplementary Data 1, James Hayden and Frederick Keeney (Imaging core at Wistar Institute) for technical supports on microscopy, Jeffrey Faust and John Fundyga (Flow cytometry core at Wistar Institute) for cell sorting, Fangping Chen (Histology core at Wistar Institute), Joel Cassel and Isabela Oliva (Screening Core at Wistar Institute) for drug resistance test, all members of the Animal Facility at The Wistar Institute and Dang lab members for their help. This work is supported by the Ludwig Institute for Cancer Research (CVD), NIH grants R01CA051497 (CVD), R01CA057341 (CVD), NIH U54-CA225088 (PKS and SS) and the Ludwig Center at Harvard (PKS and SS). S. Santagata is also supported by BWH President's Scholar Award. S.M. Pant is supported by Sigrid Juselius Foundation and Orion Research Foundation. A. Sehgal is supported by HHMI. CVD and AW are supported as Bloomberg Distinguished Professors at Johns Hopkins University.

## Author contributions

X.Z.: conceptualization, data curation, formal analysis, methodology, visualization, investigation, writing-review and editing. S.M.P.: formal analysis, visualization, validation. C.C.R.: formal analysis, methodology, visualization. H.Y.T.: data curation, formal analysis, visualization. H.S.: data curation, validation. H.D.: data curation, formal analysis, visualization. Y.Y.G.: formal analysis, visualization. R.B.: conceptualization, visualization. P.B.: data curation, formal analysis, visualization. A.J.W.: conceptualization, writing-review and editing. Y.L.: resource, writing-review and editing. A.W.: resource, writing-review and editing. A.S.: resource, writing-review and editing. M.H.: resource. A.K.: data curation, formal analysis, visualization. D.S.: methodology. P.K.S.: resource, methodology. S.S.: conceptualization, resource, writing-review and editing. C.V.D.: conceptualization, data curation, supervision, funding acquisition, writing-original draft, review and editing, project administration.

## Competing interests

The authors declare no competing interests.
