## [Peer Review File · Nature Communications]

Cell State Dependent Effects of Bmal1 on Melanoma Immunity and TumorigenicityREVIEWER COMMENTS

Reviewer #1 (Remarks to the Author): with expertise in melanoma, molecular oncology

ummary:

The authors convincingly show Bmal1 has a role in melanoma progression and anti-PD-1 immunotherapy response. They show through biotinylating assays, proximity ligation assays, and co-immunoprecipitation that Bmal1 interacts with Myh9. The authors propose this interaction sequesters Myh9 and prevents it from binding to F-actin, which allows G-actin to polymerize into F-actin and in turn, alleviate the suppression of G-actin on MRTF/SRF. The loss of suppression of MRTF/SRF leads to MRTF/SRF-driven transcription of EMT genes. They propose that this mesenchymal state ultimately leads to a more immunosuppressive tumor microenvironment that is resistant to anti-PD-1 immunotherapy. shRNA knockdown of Myh9, similar to overexpression of Bmal1, increases tumor growth rate in mice and renders tumors insensitive to anti-PD-1 immunotherapy. Targeting clock proteins and their interactors such as Bmal1 or Myh9 are potential novel therapeutic strategies.

Major comments:

It is interesting how ectopic expression of dHLH Bmal1 has a more dramatic effect in EMT signature and promotion of Sox9 expression over Sox10 as opposed to the WT. As mentioned in the manuscript, dHLH can act inhibit Clock-Bmal1 transcriptional activity, do the authors think this interaction further influences this pathway? As shown in Figure 2E, the dHLH is the ectopic protein that is driving these immunosuppressive markers, so targeting WT Bmal1 may not help increase immunogenicity in the tumor.

Is there any literature on, or do the authors speculate that manipulation of circadian clock proteins could have any negative consequences?

Minor comments:

In Figure 3, panels L and M, the colors on the images and quantifications for Sox9 and Sox10 seem to be reversed (sox9 is labeled in green in the immunofluorescence, but its

quantification is in blue where Sox10 quantification is in green).

In Figure 4, panel F, it would be interesting to quantify to determine if there is a significant difference between the WT and the dHLH, as the data so far shows more dramatic effects in the dHLH.

As mentioned in the manuscript, the role of Bmal1 in tumor progression is context dependent. Are altered EMT signatures still seen in cancers where loss of Bmal1 loss promoted tumorigenesis?

Reviewer #2 (Remarks to the Author): with expertise in Bmal1, immunology

The study by Zhang X et al examines the role of Bmal1 in melanoma tumorigenicity. Bmal1 is a master regulator of the circadian rhythm. The Dang lab has previously shown that Bmal1 gene deletion in B16 melanoma cells reduces tumor growth in C57BL/6 mice. The current work aims to understand the underlying molecular mechanism. By using Bmal1 knockout or over-expressing melanoma cells and molecular/biochemical strategies, the authors find that Bmal1 in melanoma may sequester Myh9 to promote mesenchymal cell state, immune resistant and tumorigenesis.

The authors provide ample molecular evidence to support their conclusion. However, one needs to be cautious about interpreting data derived from molecular manipulations in tumor cell lines. Notably, the authors rely heavily on the use of ectopic over-expression of Bmal1 and Bmal1 dHLH. The latter is supposed to be a dominant negative form and as such, should have the opposite activity to the WT Bmal1. However, Bmal1 dHLH was used for mechanistic studies. As such, the three components of the manuscript, Bmal1-HIF-1a crosstalk, Bmal1 dHLH and immunosuppression, and Myh9 in melanoma tumorigenesis, don't seem to have any direct connections.

Specific comments:

1. The authors did not offer any explanation as to why they switch back and forth between the YUMM2.1 wild-type and Bmal1KO cells (with/without WT or dHLH Bmal1 over-

expression). However, the concern is that the two manipulations lead to two different conclusions (see below). Although data in Fig. 1 suggest that Bmal1KO in B16 cells causes reduced HIF-1a protein and HIF-target gene expression, Bmal1 expression in WT cells doesn't really have a clear correction with HIF-1a induction by hypoxia, suggesting a secondary effect. Furthermore, Sox9 expression is independent of the HIF-1a level. There is not enough direct evidence to indicate the reduced tumor growth in Bmal1KO B16 cells is due to dampened HIF-1a activity. In Fig. 1n, the authors should include EV and mHIF-1a-TM in WT cells to assess whether mHIF-1a-TM rescues the phenotype of KO to a similar level, compared to the same manipulations in wild-type cells.

2. In Figure 2, the authors use ectopic over-expression in wild-type cells to demonstrate a role for Bmal1 in tumor-immune cell interaction. It is shown that Bmal1 dHLH tumors exert a strong immunosuppressive effect and are resistant to anti-PD1 treatment. It is unclear whether this effect is relevant as dHLH has a very weak tumor promoting effect that the EV control (Fig. 2b) and even shows a trend toward protection in the KO cell background (Fig. 1l).

3. The concern over potential artifacts caused by over-expression, notably dHLH, becomes even more apparent in the RNAseq analyses shown in Fig. 3a, which shows a substantial difference in gene expression profile in dHLH cells compared to either WT Bmal1 or EV cells. In Fig. 3b & C, the authors conclude that the comparisons between WT vs EV and dHLH vs EV demonstrate that Bmal1 shifts YUMM2.1 cells towards a mesenchymal state. However, one could argue that a more proper comparison should be WT vs dHLH (both were over-expression and dHLH serves as dominant-negative control). This comparison is likely to lead to a very different conclusion in that WT Bmal1 blocks, rather than promotes epithelial mesenchymal transition.

4. Ectopic over-expression of a transcription factor, particularly a mutated form lacking the DNA domain, is known to have squelching (or sequestration) effects. This is evident in reporter assays in Fig. 4J and extended data Fig. 4d. In addition, the effects on inducing AP-1 and bZIP ATF family (Fig. 4f/g) and epigenetic changes (Fig. 4K) are primarily seen with dHLH, but not WT Bmal1. Again, if the described mechanism is critical for the tumor-promoting effect of Bmal1, why dHLH does so little in tumor growth?

5. dHLH also causes mis-localization of dHLH to the cytosol (extended data Fig. 4a; the N-terminus DNA binding bHLH region contains NLS, Mol Cell Bio 26: 7318). This raises another

question as to whether the interaction with the predominantly cytosolic Myh9 is relevant pathologically. Most studies in Fig. 5 and 6 only compare shMyh9 in EV cells to shNC in WT or dHLH cells. To demonstrate a Myh9-dependent effect, the authors need to conduct similar experiments comparing shNC and shMyh9 in WT and dHLH cells. One would expect Myh9 knockdown in WT or dHLH cells will not exert additional tumorigenic activities.

Reviewer #3 (Remarks to the Author): with expertise in melanoma, circadian clock

Summary:

The manuscript by Zhang and Dang et al focuses on addressing the role of BMAL1 in the carcinogenic process in melanoma cancer. Although the topic is of interest, the concept that BMAL1 affects melanoma growth is not novel as Cermakian's and Castrucci's labs (see Kiessling et al., 2017; de Assis et al., 2022, see below) have previously shown evidence for this interaction. However, the authors made a great effort in further evaluating this interaction using different methods and techniques.

The authors show in two melanoma cell lines that Bmal1 KO reduces tumor growth, which is rescued partially by ectopically expression of Hifa. These effects are only seen in vivo.

Ectopic BMAL1 expression rescues tumor growth in BMAL1 KO cells as it increased immune infiltration and tumor development. Interestingly, dHLH BMAL1 cells are resistant to PD-1 treatment, which shows a remarkable effect of BMAL1 in regulating tumorigenesis. The authors demonstrate that dHLH-Bmal1 shift towards mesenchymal state and have higher and lower Sox9 and 10 expression, respectively, which resembles a human melanoma model. Interestingly, it is shown that BMAL1 interacts with Myh9, which leads to increased MRTF-SRF activity.

In summary, the authors suggest that loss of Bmal1 decreases HIF tumorigenesis while its ectopic expression, either the WT or dHLH isoform, sequesters Myh9, which regulates SRF activity. Ultimately, these changes result in mesenchymal transformation which is associated with an immune resistance phenotype.

- What are the noteworthy results?

The concept of BMAL1 regulating melanoma tumor development is not novel per se. However, the authors in this manuscript have extended this concept and provided a

possible mechanism of how BMAL1 regulates melanoma development, which is relevant and important.

- Will the work be of significance to the field and related fields? How does it compare to the established literature? If the work is not original, please provide relevant references.

The concept shown is important to the field. I would not consider the study as fully original as previous studies have shown this interaction. This study is incremental.

Unfortunately, the authors ignored pioneering studies in the field that corroborate the authors' findings. The authors should have given proper credit for the previous studies, which have already suggested a role of BMAL1 as a modulator of melanoma progression.

The authors must in the revised version acknowledge the following studies and discuss their findings considering these previous findings.

a. First report showing the role of BMAL1 in human melanoma as prognostic marker and immunotherapy biomarker (<https://pubmed.ncbi.nlm.nih.gov/29946530/>)

b. First report showing the knockout of OPN4 results in reduced tumor growth via increased immune system infiltration, decreased Mitf signaling, and increased BMAL1 expression (<https://pubmed.ncbi.nlm.nih.gov/35562405/>)

c. First report showing that BMAL1 controls the circadian expression of MITF in human melanoma cells and upregulate melanin synthesis (<https://pubmed.ncbi.nlm.nih.gov/34160901/>)

d. First report that knockdown Bmal1 results in slower tumor progression in melanoma (<https://pubmed.ncbi.nlm.nih.gov/28196531/>)

- Does the work support the conclusions and claims, or is additional evidence needed?

Throughout the manuscript the authors use a minimal n number (n = 2) for most omics approaches. Closer inspection in the supplementary files and some heatmaps for RNAseq show high variability among the samples. The combination of these factors is extremely worrying for the conclusions drawn by the authors. If the authors were focused on a circadian profile, I would be considered acceptable having a n of 2. However, having in mind the experimental setup, I consider that the study is limited to draw conclusions due to a low statistical power. The authors must clarify and clearly mention the number of independent biological samples that were used.

It is unclear the n number for WB and cytokine arrays experiments. It seems that conclusions were drawn by a n of 1. Please clarify.

- Are there any flaws in the data analysis, interpretation and conclusions? - Do these prohibit publication or require revision?

I agree with the interpretation of the data. However, as described above this study has the potential to show several false positives as in most, if not all the conditions for omics approaches, an n of 2 was used. In particular, RNAseq data was not filtered to remove lowly expressed genes, which increases the noise of the dataset. Of concern is the fact that in figure 5, the authors seem to draw their conclusion based on uncorrected p-value (not adj p value).

- Is the methodology sound? Does the work meet the expected standards in your field?

The authors should be praised for using several techniques, which allow the creation of potentially rewarding story. However, most of the omics experiments' conclusions arise from a n of 2 experiment. The authors should clearly address this either by increasing the n number for critical experiments and clearly specifying that their study is underpowered. It would be interesting if the authors could provide sample size estimation using specific programs.

- Is there enough detail provided in the methods for the work to be reproduced?

Material and method information for some techniques is poorly described while others are overdetailed.

Data deposit and availability is missing and it should be present.

The presentation of the supplementary tables is not intuitive and clear. It is hard to understand the table organization. The authors must reorganize the table formats and content. In specific cases, supplementary data for some techniques is missing (such as ChIPSeq – see comments below).

Comparisons for RNASeq are not clearly shown as it is not clear when a paired test or likelihood ratio test (LRT) in DESEQ2 was performed. Please clarify.

Specific comments.

1. Figure 1

1.1. The authors used QuantSeq 3' mRNAseq to evaluate the transcriptional changes evoked by knocking out Bmal1 in B16 cells. Closer inspection of the targeted genes shows a high variability for Hifa in Bmal1KO cells. Bmal1 gene expression seems to be slightly reduced, but BMAL1 protein knockout is clear. The authors should briefly explain this "apparent" contraction.

1.2. It is not clear which B16 subtype was used. This information is critical as different phenotypes are found among F0, F1, and F10 clones. The authors must clearly specify this in the M&M.

1.3. Table S1 lacks all comparative parameters such as fold change and adjusted p value. Please adjust.

1.4. All GSEA findings should be provided in the supplementary files. Also provide more information on how GSEA was made.

1.5. It should be clearly mentioned the n number of all WB data in addition to a proper quantification for each WB data.

1.6. RNAseq data should be filtered to remove low expressed genes and comparisons should be only based on adjust p value in DESEQ2 package.

2. Figure 2:

2.1. In figure 2A, the differences in WB data are hard to see. A quantification and the total n number for each condition is required.

2.2. In figure 2D is unclear why the authors measured cytokines in the supernatant as differences are only seen in vivo. Does the identified cytokines favor tumor growth? Are there studies from the literature to support this? The same question applies for figure 6C.

2.3. In figure 2D the n number must be provided as it seems that n of 1 was performed. Although this reviewer understands that this method is used as a screening strategy, proper validation for the targeted cytokines must be done before drawing any conclusions. In the M&M, Immunoassay method is missing.

2.4. In figure 2E gating strategy is missing and should be provided as a supplementary figure.

2.5. In figure 2E (line 138) the authors claim that NK is different between EV vs dHLH, which is not shown in the graph.

2.6. It would be interesting to evaluate the subtypes of CD4+ (naïve, central or effector

memory). The authors are invited to comment on this.

3. Figure 3:

3.1. Supplementary data for ChipSeq must be provided as a supplementary table.

3.2. In extended fig 3F, WB for SOX10 is difficult to see a difference without proper quantification.

4. Figure 4

4.1. I suggest the authors to specific nucleus and cytoplasmatic proteins in a separate sheet. GSEA and STRING analysis would also be of interest to the manuscript. This would allow an in-depth characterization of these proteins for the readers.

4.2. However, I find supressing that the Turbo ID method revealed just a handful of targets, which suggests that the method is not that specific. There are interesting candidates, but they showed in the negative control, thus impairing conclusions. The authors are invited to discuss this the manuscript.

5. Figure 5

5.1. It seems that conclusions from figure 5D were drawn by using a regular p-value and not FDR. The identity of the 925 must be shown in table S4.

5.2. More concerning is the fact that the authors should use only FDR instead of raw p value. Although, the authors in the M&M briefly mention this, it is not mentioned in the figure. In table S4, using an FDR < 0.05, a total of 270 genes are identified. Considering that the authors do not filter for lowly expressed genes, using an uncorrected p-value would result in a huge number of false positives. The authors are required to comment and address this issue.

Reviewer report

Key results

This article provides us a beautiful story about how Bmal1 change the melanoma cell status and tumorigenesis. The authors have performed some elegant experiments to prove the functional interaction between Bmal1an Myh9 which enhances MRTF-SRT activity and drives the mesenchymal transition.

Significance

These findings are very interesting which provide a clear link between circadian gene and tumorigenesis.

Data and methodology

1 It would be easier to read if put a bar graph to show the quantity of protein levels of the western blot data.

2 Interpretation of the data for why WT-bmal1 but not dHL-Bmal1 increase Sox9 expression in Bmal1-null YUMM2.1 cell (page 6 line 171-175) is not that convincing to me.

3 The results showed that WT-Bmal1 cell generated higher tumor volume than dHLH. Does that mean WT-Bmal1 has a higher Bmal1 protein level than dHLH? Why does dHLH-Bmal1 drives a more mesenchymal epigenetic state? Because it cause a stronger post-transcriptional mechanism?

Analytical approach

The analytical approach is correct.

Suggested improvements

The mechanism is clearly explained. Therefore, I don't suggest any more experiments.

Clarity and context

It would be better if the author provides some more background when it comes to the key words, for example, epithelial to mesenchymal transition and dHLH. That would be helpful for the readers to understand the whole picture.

References

The references are appropriately cited.

Your expertise

We have never used the TurboID proximity labeling system, so this part is out of the scope of my expertise.

RESPONSE TO REVIEWERS' COMMENTS

Reviewer #1 (Remarks to the Author): with expertise in melanoma, molecular oncology

Summary:

The authors convincingly show Bmal1 has a role in melanoma progression and anti-PD-1 immunotherapy response. They show through biotinylating assays, proximity ligation assays, and co-immunoprecipitation that Bmal1 interacts with Myh9. The authors propose this interaction sequesters Myh9 and prevents it from binding to F-actin, which allows G-actin to polymerize into F-actin and in turn, alleviate the suppression of G-actin on MRTF/SRF. The loss of suppression of MRTF/SRF leads to MRTF/SRF-driven transcription of EMT genes. They propose that this mesenchymal state ultimately leads to a more immunosuppressive tumor microenvironment that is resistant to anti-PD-1 immunotherapy. shRNA knockdown of Myh9, similar to overexpression of Bmal1, increases tumor growth rate in mice and renders tumors insensitive to anti-PD-1 immunotherapy. Targeting clock proteins and their interactors such as Bmal1 or Myh9 are potential novel therapeutic strategies.

RESPONSE: We thank Reviewer #1 for seeing the merit of our work and recognizing the potential contribution of our work to the literature, providing several novel concepts for further exploration in the field.

Reviewer #1 Major comments:

It is interesting how ectopic expression of dHLH Bmal1 has a more dramatic effect in EMT signature and promotion of Sox9 expression over Sox10 as opposed to the WT. As mentioned in the manuscript, dHLH can act inhibit Clock-Bmal1 transcriptional activity, do the authors think this interaction further influences this pathway? As shown in Figure 2E, the dHLH is the ectopic protein that is driving these immunosuppressive markers, so targeting WT Bmal1 may not help increase immunogenicity in the tumor.

RESPONSE: Reviewer #1 asks whether the ability of dHLH Bmal1 to inhibit the clock contributes to its immunosuppressive function in YUMM2.1 cells. To review, our data suggest that both ectopic WT and dHLH Bmal1 drives a shift toward a mesenchymal-like state associated with immunoresistance in vivo. As such, we don't believe that the inhibitor effect of dHLH Bmal1 contributes to immunosuppression per se. Rather, our results suggest that the ability of ectopic Bmal1 proteins to bind and sequester Myh9 drives toward an undifferentiated cell state drives resistance to anti-PD1 therapy. Notably, immunosuppression is phenocopied by knockdown of Myh9 (**Fig.6**), supporting the notion that sequestration of Myh9 by Bmal1 proteins plays a significant role. This cell state is associated with enhanced AP-1 transcriptional activity as seen with BRAF inhibitor resistant melanoma (PMID 28607484). Specifically, our new proximity ligation assay (PLA; revised **Fig. 4g and Supplementary Fig. 4e**) data indicate that dHLH Bmal1 associates with Myh9 and has higher PLA signals than WT Bmal1,

correlating with a more dramatic effect in EMT signature and robust resistance to anti-PD1 treatment. So, we interpret the data to surmise that the interaction between dHLH Bmal1 with Myh9 is important for the immunosuppression phenotype. With regard to targeting WT Bmal1, we surmise that if WT Bmal1 is overexpressed in resistant disease, as recently reported for enzalutamide-resistant human prostate cancer (PMID 35754340), then it is possible that reduction of Bmal1 expression could be beneficial. However, this notion remains to be further established in future studies.

Reviewer #1 Is there any literature on, or do the authors speculate that manipulation of circadian clock proteins could have any negative consequences?

RESPONSE: As mentioned, BMAL1 was recently implicated in anti-androgen therapy resistance in human prostate cancer (PMID 35754340). As such, it is possible that manipulation of clock proteins could have positive consequences for cancer. This was in part suggested by the use of a REV-ERB α (a negative regulator of BMAL1) agonist to curb glioblastoma in a mouse model (PMID 29320480).

Reviewer #1 Minor comments:

In Figure 3, panels L and M, the colors on the images and quantifications for Sox9 and Sox10 seem to be reversed (sox9 is labeled in green in the immunofluorescence, but its quantification is in blue where Sox10 quantification is in green).

RESPONSE: We thank Reviewer #1 for highlighting this. We have now corrected this (Revised **Fig. 3I**).

Reviewer #1 In Figure 4, panel F, it would be interesting to quantify to determine if there is a significant difference between the WT and the dHLH, as the data so far shows more dramatic effects in the dHLH.

RESPONSE: We thank Reviewer #1 for suggesting quantifying the data. In fact, the experiment was repeated with similar results which are now quantified. As mentioned, the quantitative PLA data (new **Fig. 4g**) indicate that dHLH Bmal1 associates with Myh9 more robustly than WT Bmal1.

Reviewer #1 As mentioned in the manuscript, the role of Bmal1 in tumor progression is context dependent. Are altered EMT signatures still seen in cancers where loss of Bmal1 loss promoted tumorigenesis?

RESPONSE: It is reported in the literature that loss of Bmal1 in colon carcinoma (CRC) cells resulted in a mesenchymal to epithelial shift (MET) and diminished chemoresistance (PMID 34065633). This study did not perform in vivo tumorigenesis experiments. Conversely, Bmal1 overexpression increased in vivo tumorigenesis and induced EMT in CRC cells (PMID 36806631). By contrast, another study in GEMM reported that loss of Bmal1 accelerated Apc-driven CRC pathogenesis (PMID 35947664) with increased EMT. The fact that overexpression (PMID 36806631) or loss of Bmal1 (PMID 35947664) can be associated with EMT suggests a dependency on context. Whether the contextual difference here depends on the model, ie. tumor initiation with the GEMM Apc loss model vs. tumor progression with established tumors, is unclear. We agree, however, the role of Bmal1 in tumorigenesis is context dependent as stipulated by our study.

Reviewer #2 (Remarks to the Author): with expertise in Bmal1, immunology

Reviewer #2 The study by Zhang X et al examines the role of Bmal1 in melanoma tumorigenicity. Bmal1 is a master regulator of the circadian rhythm. The Dang lab has previously shown that Bmal1 gene deletion in B16 melanoma cells reduces tumor growth in C57BL/6 mice. The current work aims to understand the underlying molecular mechanism. By using Bmal1 knockout or over-expressing melanoma cells and molecular/biochemical strategies, the authors find that Bmal1 in melanoma may sequester Myh9 to promote mesenchymal cell state, immune resistant and tumorigenesis.

The authors provide ample molecular evidence to support their conclusion. However, one needs to be cautious about interpreting data derived from molecular manipulations in tumor cell lines. Notably, the authors rely heavily on the use of ectopic over-expression of Bmal1 and Bmal1 dHLH. The latter is supposed to be a dominant negative form and as such, should have the opposite activity to the WT Bmal1. However, Bmal1 dHLH was used for mechanistic studies. As such, the three components of the manuscript, Bmal1-HIF-1a crosstalk, Bmal1 dHLH and immunosuppression, and Myh9 in melanoma tumorigenesis, don't seem to have any direct connections.

RESPONSE: We appreciate Reviewer #2's observation that we have provided significant amounts of evidence to support our conclusion. We wish to address the appearance of lack of connections between Bmal1-HIF crosstalk, Bmal1 dHLH and immunosuppression and interaction with Myh9. We regret that we were not clearer in the original manuscript and hope that the revision clarifies the connections.

The intention of our study was to determine the consequences of loss of Bmal1 versus ectopic Bmal1 expression on melanoma tumorigenesis. As per our report, we found that loss of endogenous Bmal1 in murine melanoma cells accompanies a decrease in expression of HIF-1 targets through unbiased analysis of transcriptomic changes. However, upon ectopic expression of WT Bmal1 or dHLH Bmal1 in parental cells, we were surprised to observe not only increased tumorigenesis, but also an immune-evasive phenotype with resistance to anti-PD1 therapy. Changes in the transcriptome by WT and dHLH Bmal1 indicate a shift toward an undifferentiated mesenchymal state with reduced Sox10 and increased Sox9 expression. These observations, particularly the finding that tumors with WT Bmal1 and dHLH Bmal1 progressively increased immune evasion (more resistant to anti-PD1 therapy), led us to hypothesize that it is not the dominant negative function of dHLH Bmal1 (which would be expected to reverse the phenotype of WT Bmal1 if dHLH were to be dominant negative), but a non-transcriptional function that conferred this effect. As such, we sought in an unbiased fashion, using TurboID, proteins that may interact with Bmal1 or dHLH Bmal1 and found Myh9 as a major interactor. Further, we document that loss of Myh9 phenocopies ectopic expression of WT and dHLH Bmal1 with respect to anti-PD1 resistance. Hence, the connections between the different aspects of this manuscript are established.

We also wish to address specifically the notion of dHLH Bmal1's activity as one of a dominant negative nature. Indeed, while dHLH Bmal1 has a dominant negative transcriptional effect on Bmal1 target genes (**Supplementary Fig. 2b**), its non-transcriptional function of interacting with Myh9, however, appears to play a significant role in its induction of the mesenchymal cell state and immune evasion in vivo. Further, it should be noted that immune evasion does not necessarily lead to larger tumors but does lead to tumors that resist anti-PD1 therapy.

Reviewer #2 Specific comments:

Reviewer #2 1. The authors did not offer any explanation as to why they switch back and forth between the YUMM2.1 wild-type and Bmal1KO cells (with/without WT or dHLH Bmal1 over-expression). However, the concern is that the two manipulations lead to two different conclusions (see below).

RESPONSE: We apologize for not making our observations clearer, causing the appearance of switching between topics as well as confusion over loss of Bmal1 function and Bmal1 ectopic expression. Please note as mentioned above the difference between loss of endogenous Bmal1 and effects of ectopic expression of WT or dHLH Bmal1. Indeed, the gain or loss of Bmal1 function is context dependent and not necessarily the converse of one another; ie, loss of Bmal1 should have a diametrically opposite effect of Bmal1 overexpression. Here, it should be noted that an increase in Bmal1 expression and function appears to underlie the ability of human prostate cancer to resist enzalutamide therapy, whereas endogenous Bmal1 is insufficient to confer resistance (PMID 35754340). This prostate cancer study illustrates the differences between endogenous Bmal1 versus increased Bmal1 neomorphic activity in human tumor biology. Hence, loss of Bmal1 in YUMM2.1 yielded a phenotype that is not the converse of ectopic expression of Bmal1. We hope that the explanation above connects the different concepts and clarifies the issue.

Reviewer #2 Although data in Fig. 1 suggest that Bmal1KO in B16 cells causes reduced HIF-1 α protein and HIF-target gene expression, Bmal1 expression in WT cells doesn't really have a clear correction with HIF-1 α induction by hypoxia, suggesting a secondary effect. Furthermore, Sox9 expression is independent of the HIF-1 α level. There is not enough direct evidence to indicate the reduced tumor growth in Bmal1KO B16 cells is due to dampened HIF-1 α activity. In Fig. 1n, the authors should include EV and mHIF-1 α -TM in WT cells to assess whether mHIF-1 α -TM rescues the phenotype of KO to a similar level, compared to the same manipulations in wild-type cells.

RESPONSE: As pointed out above, loss of Bmal1 and gain of ectopic Bmal1 are not the converse of one another. We believe that the recent findings in human prostate cancer (PMID 35754340) illustrates that gain of Bmal1 function confers enzalutamide resistance, a phenotype that is not conferred by endogenous Bmal1. In parallel, we find that loss of Bmal1 was associated with decreased expression of HIF-1 target genes, but ectopic expression of Bmal1 or dHLH Bmal1 conferred a different phenotype of EMT and immune evasion through Bmal1 proteins interaction with Myh9.

We document that Sox9 expression parallels that of HIF-1 α but did not mean to infer a cause-and-effect connection. To test the hypothesis that HIF-1 α may play a role in decreased tumorigenesis of Bmal1 knockout YUMM2.1 cells, we expressed a stabilized HIF-1 α (TM) which rescued growth (**Fig. 1n**). This observation is consistent with HIF-1 α decreased activity in YUMM2.1 KO cells contributed to its slowed in vivo growth. Further, we now showed that ectopic expression of HIF-1 α -TM also modestly although not significantly increased tumorigenesis in WT cells (**new Supplementary Fig. 1k**) as might be expected from our earlier studies showing that HIF-1 α -TM can increase tumorigenesis, presumably by providing better adaptation to in vivo hypoxia (PMID 17785204). Overall, we suggest that loss of Bmal1 diminishes tumorigenesis at least partially through reduction in Hif1 α activity.

To provide direct evidence to indicate the reduced tumor growth in Bmal1KO B16 cells is due to dampened Hif1 α activity, we tried with significant time dedication to overexpress stabilized Hif1 α .

(Hif1 α -TM) in Bmal1KO B16 cells, but these cells do not tolerate persistent Hif1 α -TM expression for reasons that we don't understand. We assessed directly the tolerance of Bmal1KO B16 cells to stabilized Hif1 α expression through time course experiments. Please see details below:

Experiment: B16 Bmal1 KO cells were transduced with lentiviruses expressing GFP and Hif1 α -TM. Empty vector only expressing GFP was used as a control. We could not retrieve B16 Bmal1 KO cells stably expressing Hif1 α -TM. We then sought to determine the expression of Hif1 α -TM in a time course experiment. The percentage of GFP+ cells were test by flow cytometry every two day and the cell lysate was made at the corresponding time to measure Hif1 α -TM protein level. We found that both GFP and Hif1 α -TM level diminished over 8 days (**Data for cells without GFP sorting**). In addition, we also sorted the GFP positive cells on day 2 after virus transduction and then followed Hif1 α -TM expression levels every three days. As with unsorted cells, the percentage of GFP and Hif1 α -TM in sorted cells decreased over time (**Data for cells after GFP sorting**). Because of the inability to expressed stabilized Hif1a in the B16 Bmal1 KO cells, we are unable to study these cells in vivo. In this respect, we have modified our statement regarding the role of HIF-1 α in B16 Bmal1 KO cells, notwithstanding the transcriptomic changes corresponding to decreased HIF-1 target gene expression.

Cells after GFP sorting

Meanwhile, different from B16 Bmal1 KO cells, B16 WT cells tolerated Hif1α-TM. Our in vivo experiment with Hif1α-TM overexpression in B16 WT cells showed similar result as seen in YUMM2.1 WT cells. Please see below (for reviewer only):

Reviewer #2: 2. In Figure 2, the authors use ectopic over-expression in wild-type cells to demonstrate a role for Bmal1 in tumor-immune cell interaction. It is shown that Bmal1 dHLH tumors exert a strong

immunosuppressive effect and are resistant to anti-PD1 treatment. It is unclear whether this effect is relevant as dHLH has a very weak tumor promoting effect that the EV control (Fig. 2b) and even shows a trend toward protection in the KO cell background (Fig. 1l).

RESPONSE: We wish to clarify our observations. We believe that there may be some confusion about the notion of immunoresistance and tumor promoting effect (ie, overall tumor growth rates). It stands to reason that the overall tumor growth rate depends on: a) cell intrinsic proliferation rate (%Ki67 or tumor growth fraction), b) ability to resist cell death due to non-immune metabolic environment and c) ability to resist anti-tumor immunity. As such, dHLH confers immunoresistance as documented in our study using anti-PD1 treatment. The WT Bmal1 tumors, however, grew faster than dHLH tumors. We interpret these findings to mean that WT Bmal1 tumors have an intrinsic growth advantage along with immunoresistance, whereas dHLH tumors are highly immunoresistant but have decreased tumor cell intrinsic growth in vivo. We surmise that the decrease in Sox10 expression, which is associated with melanoma tumor growth rate (PMID: 34879275), in dHLH Bmal1 tumors contributes to their slower cell intrinsic growth rate in vivo. We performed Ki67+ staining for melanoma cells in the tumor tissues and found that the percentage of Ki67+ melanoma cells in dHLH tumors are lower than both WT and EV tumors (new **Supplementary Figs. 2g and h**), which means a lower cell intrinsic growth rate. Lastly, we don't believe that the data in Fig 1l illustrates a trend toward protection of KO cells by dHLH Bmal1.

Reviewer #2: 3. The concern over potential artifacts caused by over-expression, notably dHLH, becomes even more apparent in the RNAseq analyses shown in Fig. 3a, which shows a substantial difference in gene expression profile in dHLH cells compared to either WT Bmal1 or EV cells. In Fig. 3b & C, the authors conclude that the comparisons between WT vs EV and dHLH vs EV demonstrate that Bmal1 shifts YUMM2.1 cells towards a mesenchymal state. However, one could argue that a more proper comparison should be WT vs dHLH (both were over-expression and dHLH serves as dominant-negative control). This comparison is likely to lead to a very different conclusion in that WT Bmal1 blocks, rather than promotes epithelial mesenchymal transition.

RESPONSE: In principle, we agree with Reviewer #2 that ectopic expression of WT or dHLH Bmal1 is an artificial experimental construct which was designed to probe a tumor system in response to Bmal1 manipulation. In fact, we chose this approach to ask how overexpressed Bmal1, as now found in human prostate cancer resistant to enzalutamide (PMID 35754340), affects tumorigenesis. Indeed, Reviewer #2 is correct that dHLH Bmal1 confers a stronger shift toward the mesenchymal state associated with higher AP1 activity and markedly decreased Sox10 expression.

We respectfully disagree, however, that dHLH must function in opposition to Bmal1 as a dominant negative allele *a priori* in light of evidence that both ectopic dHLH and WT Bmal1 proteins interact with Myh9. Further, the manipulation of the system is from basal state with endogenous Bmal1 (EV) to two altered states with WT Bmal1 and dHLH Bmal1 overexpression. As such, we respectfully disagree that comparing WT Bmal1 to dHLH Bmal1 is appropriate without considering the basal state of endogenous Bmal1. This contorted comparison would lead to a distorted conclusion that WT Bmal1 blocks EMT, particularly in light of the fact that WT Bmal1 shifts the basal EV state toward EMT rather than suppressing it.

Reviewer #2: 4. Ectopic over-expression of a transcription factor, particularly a mutated form lacking the

DNA domain, is known to have squelching (or sequestration) effects. This is evident in reporter assays in Fig. 4J and Supplementary Fig. 4d. In addition, the effects on inducing AP-1 and bZIP ATF family (Fig. 4f/g) and epigenetic changes (Fig. 4K) are primarily seen with dHLH, but not WT Bmal1. Again, if the described mechanism is critical for the tumor-promoting effect of Bmal1, why dHLH does so little in tumor growth?

RESPONSE: Here, we believe that there is confusion over the difference between cell states associated with immune evasion and intrinsic tumor growth rates that depends on growth fraction as discussed above. In this regard, new **Supplementary Figs. 2g and h** indicate that Ki67⁺ is higher in WT versus dHLH Bmal1 tumors, indicating a higher growth fraction in WT tumors. There appears to be confusion with notion that dHLH Bmal1 must function as a dominant negative transcriptional allele, particular since we document that WT Bmal1 and dHLH Bmal1 can both interact with Myh9 in a non-transcriptional role; a role that is importantly phenocopied by decreased Myh9 expression. It is notable that this is not unlike moonlighting metabolic enzymes in the nucleus that play previously unforeseen non-catalytic roles (PMID 5752986); likewise, the oncogenic transcription factor MYC also has non-transcriptional roles (PMID: 36424410; PMID: 24681440). Hence, we believe that the preconceived notion of expected functions may not hold up to experimental findings.

Reviewer #2: 5. dHLH also causes mis-localization of dHLH to the cytosol (Supplementary Fig. 4a; the N-terminus DNA binding bHLH region contains NLS, Mol Cell Bio 26: 7318). This raises another question as to whether the interaction with the predominantly cytosolic Myh9 is relevant pathologically. Most studies in Fig. 5 and 6 only compare shMyh9 in EV cells to shNC in WT or dHLH cells. To demonstrate a Myh9-dependent effect, the authors need to conduct similar experiments comparing shNC and shMyh9 in WT and dHLH cells. One would expect Myh9 knockdown in WT or dHLH cells will not exert additional tumorigenic activities.

RESPONSE: We acknowledge that dHLH is found in the cytosol, but as indicated by data shown in **Fig. 4f** and **Supplementary Fig. 4d**, the interaction between Bmal1 and Myh9 are mostly confined to the nucleus. Hence, the inference that dHLH mislocalized to the cytosol and must predominantly function in the cytosol does not match with the experimental findings that dHLH is largely nuclear. Notably, we document that Myh9 knockdown phenocopies ectopic expression of WT or dHLH Bmal1, providing additional evidence for a functional interaction between Bmal1 and Myh9.

Reviewer #2 surmises that Myh9 knockdown in WT cells will not exert additional tumorigenic activities, assuming *a priori* that WT Bmal1 must sequester all of Myh9. However, in the new **Supplementary Figs. 6a and b**, we document that Myh9 knockdown in WT Bmal1 cells exert modest increase in tumorigenesis, suggesting that ectopic expression of WT Bmal1 does not sequester all Myh9. Hence, it does not stand to reason that Myh9 knockdown in WT cells will not exert additional tumorigenesis activities unless WT Bmal1 sequesters all Myh9.

Reviewer #3 (Remarks to the Author): with expertise in melanoma, circadian clock

Summary:

Reviewer #3. The manuscript by Zhang and Dang et al focuses on addressing the role of BMAL1 in the carcinogenic process in melanoma cancer. Although the topic is of interest, the concept that BMAL1 affects melanoma growth is not novel as Cermakian's and Castrucci's labs (see Kiessling et al., 2017; de Assis et al., 2022, see below) have previously shown evidence for this interaction. However, the authors made a great effort in further evaluating this interaction using different methods and techniques.

RESPONSE: We appreciate Reviewer #3 noting that we have made a great effort to study the role of Bmal1 using different methods and techniques. Reviewer #3 is correct that BMAL1 manipulation has been documented to affect melanoma growth. Keissling et al. (PMID 28196531; 2017) reported that dexamethasone, which can synchronize cellular clocks in vitro can curb B16 melanoma tumor growth in both C57BL/6 and NSG mice, suggesting that the effect is independent of an intact immune system. They proceeded to demonstrate that shRNA-mediated reduction of Bmal1 rescued B16 tumor growth in the presence of dexamethasone (although the host, NSG vs C57/BL6, was not stated). Notably, however, shBmal1 (phosphate buffered saline; PBS) did not affect tumor growth compared to scramble shRNA (PBS) (see Fig.5 in PMID 28196531), contrasting with our finding that CRISPR/Cas9-mediated Bmal1 KO decreased YUMM2.1 tumor growth. The Assis et al. 2022 (PMID: 35562405) paper documents that melanopsin (Opn4) loss increased Bmal1 expression, impaired clock function in vitro, reduced MITF expression, reduced melanin content, and diminished B16-F10 tumor growth with associated changes in tumor immune cell infiltration. These studies, which are now cited and discussed. Our study further evaluated the role of Bmal1 and offers a novel perspective on context-dependent role of Bmal1 in melanoma tumorigenesis documenting that decreased Bmal1 reduced tumor growth in two different melanoma cell lines. Further, the effects of ectopic Bmal1 expression on the immune response is documented with the unforeseen interaction between Bmal1 and Myh9.

Reviewer #3. The authors show in two melanoma cell lines that Bmal1 KO reduces tumor growth, which is rescued partially by ectopically expression of Hifa. These effects are only seen in vivo. Ectopic BMAL1 expression rescues tumor growth in BMAL1 KO cells as it increased immune infiltration and tumor development. Interestingly, dHLH BMAL1 cells are resistant to PD-1 treatment, which shows a remarkable effect of BMAL1 in regulating tumorigenesis. The authors demonstrate that dHLH-Bmal1 shift towards mesenchymal state and have higher and lower Sox9 and 10 expression, respectively, which resembles a human melanoma model. Interestingly, it is shown that BMAL1 interacts with Myh9, which leads to increased MRTF-SRF activity.

In summary, the authors suggest that loss of Bmal1 decreases HIF tumorigenesis while its ectopic expression, either the WT or dHLH isoform, sequesters Myh9, which regulates SRF activity. Ultimately, these changes result in mesenchymal transformation which is associated with an immune resistance phenotype.

- What are the noteworthy results?

The concept of BMAL1 regulating melanoma tumor development is not novel per se. However, the authors in this manuscript have extended this concept and provided a possible mechanism of how BMAL1 regulates melanoma development, which is relevant and important.

RESPONSE: We thank Reviewer #3 for recognizing that our study offers a probable, important mechanism of how Bmal1 regulates melanoma development. This will allow the field to expand on this framework.

Reviewer #3. Will the work be of significance to the field and related fields? How does it compare to the established literature? If the work is not original, please provide relevant references. The concept shown is important to the field. I would not consider the study as fully original as previous studies have shown this interaction. This study is incremental.

RESPONSE: We believe that Reviewer #3's remark "I would not consider the study as fully original as previous studies have shown this interaction" refers to the roles of Bmal1 and melanoma tumorigenesis. Given the conflicting and confusion role of Bmal1 in melanoma tumorigenesis (see PMID 28196531 versus 35562405) as discussed above, we believe that a conceptual framework is lacking in the field that is fraught with preconceived notions. Reviewer #3 does not consider "the study as fully original as previous have shown this interaction." It is unclear what Reviewer #3 means by "this interaction." Indeed, earlier studies on Bmal1 and melanoma have not uncovered an interaction with HIF-1a or the binding of Bmal1 proteins with Myh9. In fact, the mechanistic understanding of how Bmal1 affects melanoma tumorigenesis is not clearly understood.

As yet, no comprehensive study on a syngeneic model melanoma has been reported that manipulated Bmal1 through CRISPR/Cas9 gene editing and ectopically expressed Bmal1 proteins, coupled with proteomic and genomic studies, to uncover the role of Bmal1 in affecting HIF1 signaling and melanoma cell states. Further, overexpression of Bmal1 confers immunotherapy resistance. This observation parallels with the recent finding that human BMAL1, downstream of FOXO A1, is elevated and necessary for enzulutamide resistance (PMID 35754340). As such, trivializing our comprehensive study as incremental is unjustified.

Reviewer #3. Unfortunately, the authors ignored pioneering studies in the field that corroborate the authors' findings. The authors should have given proper credit for the previous studies, which have already suggested a role of BMAL1 as a modulator of melanoma progression. The authors must in the revised version acknowledge the following studies and discuss their findings considering these previous findings.

- a. First report showing the role of BMAL1 in human melanoma as prognostic marker and immunotherapy biomarker (<https://pubmed.ncbi.nlm.nih.gov/29946530/>)
- b. First report showing the knockout of OPN4 results in reduced tumor growth via increased immune system infiltration, decreased Mitf signaling, and increased BMAL1 expression (<https://pubmed.ncbi.nlm.nih.gov/35562405/>)
- c. First report showing that BMAL1 controls the circadian expression of MITF in human melanoma cells and upregulate melanin synthesis (<https://pubmed.ncbi.nlm.nih.gov/34160901/>)
- d. First report that knockdown Bmal1 results in slower tumor progression in melanoma (<https://pubmed.ncbi.nlm.nih.gov/28196531/>)

RESPONSE: We regret that we did not comprehensively cite many parts of the literature on BMAL1 and melanoma. We have now discussed and included all these references.

Reviewer #3. Does the work support the conclusions and claims, or is additional evidence needed? Throughout the manuscript the authors use a minimal n number (n = 2) for most omics approaches. Closer inspection in the supplementary files and some heatmaps for RNAseq show high variability among the samples. The combination of these factors is extremely worrying for the conclusions drawn by the authors. If the authors were focused on a circadian profile, I would be considered acceptable having a n of 2. However, having in mind the experimental setup, I consider that the study is limited to draw conclusions due to a low statistical power. The authors must clarify and clearly mention the number of independent biological samples that were used. It is unclear the n number for WB and cytokine arrays experiments. It seems that conclusions were drawn by a n of 1. Please clarify.

RESPONSE: We acknowledge that we used biological duplicates is underpowered for all RNA-seq and CHIP-seq data. Further, in the supplementary RNAseq files (**Supplementary Table 1, Table 2 and Table 4**), “normalized counts” instead of log-transformed number was used to display the expression level, so higher variability is observed among the samples. We agree that the sample size will affect the statistical power, and we also know that statistical significance does not mean biological significance. So, our conclusions are drawn not only based on the statistical analyses of RNAseq or CHIP-seq, but also through several different techniques (use of proteomics and reporter assays) that allow us to draw conclusion that encompasses data from different approaches. In addition, according to Reviewer #3’s suggestion, we now clarified the number of independent biological samples in the figure legends if not previously done. Please see: **Fig.1c,d,g, Supplementary Fig. 1b, Fig. 2a, 2d, Fig. 3g, Supplementary Fig. 3g,h and Fig. 5f,i.**

Reviewer #3. Are there any flaws in the data analysis, interpretation and conclusions? - Do these prohibit publication or require revision?

I agree with the interpretation of the data. However, as described above this study has the potential to show several false positives as in most, if not all the conditions for omics approaches, an n of 2 was used. In particularly, RNAseq data was not filtered to remove lowly expressed genes, which increases the noise of the dataset. Of concern is the fact that in figure 5, the authors seem to draw their conclusion based on uncorrected p-value (not adj p value).

RESPONSE: We thank Reviewer#3 for agreeing with our interpretation of the data. And we agree that lowly expressed genes should be removed for the analysis. We have now re-analyzed our RNAseq data, better defined our criteria for analysis, and have **revised Fig.5** accordingly.

Reviewer #3. Is the methodology sound? Does the work meet the expected standards in your field? The authors should be praised for using several techniques, which allow the creation of potentially rewarding story. However, most of the omics experiments’ conclusions arise from a n of 2 experiment. The authors should clearly address this either by increasing the n number for critical experiments and clearly specifying that their study is underpowered. It would be interesting if the authors could provide sample size estimation using specific programs.

RESPONSE: We thank Reviewer #3 for praising the use of several techniques in our work and assessing that our work may generate a rewarding story, one we believe that will offer additional perspectives on the function of Bmal1 in a murine melanoma model. We recognize and now state that our study is

underpowered due to $n = 2$ instead of having more replicates. As Reviewer #3 recognize our studies are complex and resources to perform many replicates are limited, particularly if one looks at the number of studies in vitro and in vivo that have been done for our manuscript. Please note that for animal studies, we generated sample size estimate (power calculation) as per of the experimental design for approval by the institutional animal use committee.

Reviewer #3. - Is there enough detail provided in the methods for the work to be reproduced? Material and method information for some techniques is poorly described while others are overdetailed.

RESPONSE: We have reviewed our methods carefully. Where necessary, we have provided additional details. Of course, we are open to inquiries from readers who may be interested in any of our methods and will share any protocol details for replication upon request.

Reviewer #3. Data deposit and availability is missing and it should be present.

RESPONSE: At the time of submission, we deposited all omics data in appropriate publicly accessible databases. The accession numbers are now available and provided in the revised manuscript.

Reviewer #3. The presentation of the supplementary tables is not intuitive and clear. It is hard to understand the table organization. The authors must reorganize the table formats and content. In specific cases, supplementary data for some techniques is missing (such ChipSeq – see comments below).

RESPONSE: We have now provided additional details for the supplementary tables and provided extra tabs (**Supplementary Table 3**) where necessary to make the tables more useful for readers.

Reviewer #3. Comparisons for RNASeq are not clearly shown as it is not clear when a paired test or likelihood ratio test (LRT) in DESeq2 was performed. Please clarify.

RESPONSE: Instead of DESeq2 used to estimate significance of differential expression between groups of samples, DESeq2 **with default Wald test option** was used to estimate significance of differential expression between groups of samples.

Specific comments.

1. Figure 1

Reviewer #3. 1.1. The authors used QuantSeq 3' mRNAseq to evaluate the transcriptional changes evoked by knocking out Bmal1 in B16 cells. Closer inspection of the targeted genes shows a high variability for Hif1a in Bmal1KO cells. Bmal1 gene expression seems to be slightly reduced, but BMAL1 protein knockout is clear. The authors should briefly explain this “apparent” contraction.

RESPONSE: Bmal1 was knocked out by CRISPR/Cas9 system in which Cas9 specifically targets and binds to Bmal1 gene through guide RNA (gRNA; see diagram) and cuts the double strands DNA of the Bmal1 gene locus. The DNA cleavage will be repaired by non-homologous end joining (NHEJ) which results in insertions and deletion mutation of Bmal1 gene, often leading to mutant RNAs that are not translated. In this respect, a mutated RNA (say, with a mutant stop codon introduced in the coding sequence) would still be expressed but not the protein.

Reviewer #3. 1.2. It is not clear which B16 subtype was used. This information is critical as different phenotypes are found among F0, F1, and F10 clones. The authors must clearly specify this in the M&M.

RESPONSE: We thank the Reviewer#3 for pointing out the detail we missed in our previous manuscript. We clarified that the cell line used is B16-F10 in our revised manuscript.

Reviewer #3. 1.3. Table S1 lacks all comparative parameters such as fold change and adjusted p value. Please adjust.

RESPONSE: These comparative parameters are included in **Supplementary Table 1** now.

Reviewer #3. 1.4. All GSEA findings should be provided in the supplementary files. Also provide more information on how GSEA was made.

RESPONSE: All GSEA findings are provided in the supplementary files (**see Supplementary Fig. 1a, 1f, 1g, Supplementary Fig. 3b, 3c and Supplementary Fig. 5c**) and information on how GSEA was performed is described in the Methods section.

Reviewer #3. 1.5. It should be clearly mentioned the n number of all WB data in addition to a proper quantification for each WB data.

RESPONSE: We apologize for missing the n number in our figure legend before. As mentioned above, it is clarified now. Please see the figure legend for **Fig.1c, 1d, 1g, Fig. 2a, Extended Fig. 1a, Fig. 3g and Fig. 5f, 5i**. We also have now provided the quantitation of western blot data for some proteins that are not easy to tell the difference across samples by eyes as numbers below the panels. Please see **Fig. 3g, Supplementary Fig. 3g, h, Supplementary Fig. 4g, Fig. 5a, f and i**.

Reviewer #3. 1.6. RNAseq data should be filtered to remove low expressed genes and comparisons should be only based on adjust p value in DESEQ2 package.

RESPONSE: We thank Reviewer#3 for pointing out that low expressed genes should be removed for the RNAseq data analysis. Gene expression level in our RNAseq data was displayed with “normalized counts”, so the genes with counts <10 in all compared samples were removed. And comparisons are now performed based on FDR (adjust p value) instead of raw p value. It is notable that conclusions in

subsections of the manuscript are unaltered with more stringent re-analysis of the data.

2. Figure 2:

Reviewer #3. 2.1. In figure 2A, the differences in WB data are hard to see. A quantification and the total n number for each condition is required.

RESPONSE: We assume Reviewer#3 is asking about the quantification of Hif1 α protein level across all conditions. Unexpectedly, although Hif1 α was clearly diminished in Bmal1 KO cells, it was not consistently altered in YUMM2.1-WT or YUMM2.1-dHLH cells across the time course (**Fig. 2a**). Hence, we did not provide the quantification which would further complicate the figure.

Reviewer #3. 2.2. In figure 2D is unclear why the authors measured cytokines in the supernatant as differences are only seen in vivo. Does the identified cytokines favor tumor growth? Are there studies from the literature to support this? The same question applies for figure 6C.

RESPONSE: As Reviewer #3 appreciates, measurement of cytokines produced by the cancer cell lines could only be definitive in vitro because other cells in vivo could also release these cytokines. Specifically, since we found immune-evasive phenotypes of WT and dHLH-Bmal1 tumors, we sought to determine whether cytokines could be released by tumor cells that affect the tumor microenvironmental immune cells. Based on the literature, many of the cytokines or chemokines identified do favor tumor growth of different cancer types (PMID 24263190, PMID 35865534, PMID 28716888, PMID 35725813, PMID 22770218). For example, Csf1 is known to favor immune evasion through stimulation of TAMs (PMID 30718830). Because we did not specifically probe the roles of different chemokines, we refrained from speculating and hope that the data will lead to future studies in the field.

Reviewer #3. 2.3. In figure 2D the n number must be provided as it seems that n of 1 was performed. Although this reviewer understands that this method is used as a screening strategy, proper validation for the targeted cytokines must be done before drawing any conclusions. In the M&M, Immunoassay method is missing.

RESPONSE: We apologize that we just plotted the average and missing the error bar here. We corrected the graph by adding error bar and appropriate statistical analysis. The cytokines measurement was done by RayBiotech. All samples were prepared according to the instruction, and medium without cells were used as a control. Please see here for details: <https://www.raybiotech.com/mouse-cytokine-array-q1000-qam-caa-1000>.

Reviewer #3. 2.4. In figure 2E gating strategy is missing and should be provided as a supplementary figure.

RESPONSE: Gating strategy is now included in the Supplementary Fig. 2i.

Reviewer #3. 2.5. In figure 2E (line 138) the authors claim that NK is different between EV vs dHLH, which is not shown in the graph.

RESPONSE: We thank Reviewer#3 for pointing out our confusing description here. It is revised now.

Reviewer #3. 2.6. It would be interesting to evaluate the subtypes of CD4+ (naïve, central or effector memory). The authors are invited to comment on this.

RESPONSE: We recognize that CD4+ T cells are significantly increased in WT and dHLH tumors. And we thank the suggestion of Reviewer#3, but we believe that further studies on the CD4+ subtypes are beyond the scope of the current, already complex, study. Here, we want to reveal if WT or dHLH-Bmal1 overexpression affects immune cells composition in the tumors, and we found MDSCs and Macrophages that play very important role in tumor immune response are significantly changed.

3. Figure 3:

Reviewer #3. 3.1. Supplementary data for ChipSeq must be provided as a supplementary table.

RESPONSE: All raw data for ChIPseq are available in GEO for downloading. We believe that the manuscript is already very large in content and hence, tables can be available upon request.

Reviewer #3. 3.2. In extended fig 3F, WB for SOX10 is difficult to see a difference without proper quantification.

RESPONSE: The quantification is completed and included in the extended Fig. 3e now.

4. Figure 4

Reviewer #3. 4.1. I suggest the authors to specific nucleus and cytoplasmic proteins in a separate sheet. GSEA and STRING analysis would also be of interest to the manuscript. This would allow an in-depth characterization of these proteins for the readers.

RESPONSE: We thank Reviewer#3 for the suggestion to specify nucleus and cytoplasmic proteins in a separate sheet. As we know that lots of proteins are shuttle proteins that not only exist in nuclear but also in cytoplasm, so, here we used the ratio of protein intensity in Tb to TbNLS to distinguish these proteins and listed them in separate sheets in Table S3. We believe that additional STRING analysis will make our manuscript more cumbersome and distract from the major thrust of our work. In this respect, our omics data are available for further analyses.

Reviewer #3. 4.2. However, I find suppressing that the Turbo ID method revealed just a handful of targets, which suggests that the method is not that specific. There are interesting candidates, but they showed in the negative control, thus impairing conclusions. The authors are invited to discuss this the manuscript.

RESPONSE: Respectfully, we believe that the *a priori* notion that labeling of a few targets suggest lack of specificity is tautological and not based on the experimental facts. The fact is that we used proper controls in our TurboID experiment with Tb, TbNLS and No biotin treatment to exclude the non-specific proteins enriched by Bmal1-Tb fusion proteins. In fact, it stands to reason that if our labeling is pervasive

and labels many proteins, then the counter argument that it must be non-specific. Importantly, we wish to emphasize that our controls allow us to definitively show specificity over background, signal over noise. We believe that we have already discussed why we used the Tb and Tb-NLS controls (which tend to be lacking in the literature) in the manuscript.

5. Figure 5

Reviewer #3. 5.1. It seems that conclusions from figure 5D were drawn by using a regular p-value and not FDR. The identity of the 925 must be shown in table S4.

RESPONSE: We thank Reviewer#3 for pointing this out. As mentioned above, we re-analyzed the data and used FDR instead of raw of p value to identify the overlaps of these significantly changed genes (627 genes which include 413 upregulated genes and 214 downregulated genes, revised **Fig. 5d**) and the conclusions remains the same. Also, 413 genes significantly increased by shMyh9, WT-Bmal1 and dHLH-Bmal1 are shown in Table S4. The overall conclusion did not change upon more stringent re-analysis of the data.

Reviewer #3. 5.2. More concerning is the fact that the authors should use only FDR instead of raw p value. Although, the authors in the M&M briefly mention this, it is not mentioned in the figure. In table S4, using an FDR < 0.05, a total of 270 genes are identified. Considering that the authors do not filter for lowly expressed genes, using an uncorrected p-value would result in a huge number of false positives. The authors are required to comment and address this issue.

RESPONSE: As mentioned above, all these issues are addressed in our revised manuscript. We removed lowly expressed genes for RNA-seq data analysis, used FDR instead of raw p value. We apologize that Table S4 was not more explicit; we have corrected it, now showing a total of 627 genes identified using FDR < 0.05 (revised **Fig. 5d**). The corresponding figures are also revised with the new genes list.

Reviewer #4

Key results

This article provides us a beautiful story about how Bmal1 change the melanoma cell status and tumorigenesis. The authors have performed some elegant experiments to prove the functional interaction between Bmal1 and Myh9 which enhances MRTF-SRT activity and drives the mesenchymal transition.

RESPONSE: We thank Reviewer #4 appreciating the quality of our work and for recognizing the key message of our work that Bmal1 overexpression induces a cell state switch that depends on the unforeseen Bmal1-Myh9 interaction, driving MRTF-SRF and AP1 activities.

Reviewer #4 Significance

These findings are very interesting which provide a clear link between circadian gene and tumorigenesis.

RESPONSE: We thank Reviewer #4 for this comment.

Data and methodology

Reviewer #4 1 It would be easier to read if put a bar graph to show the quantity of protein levels of the western blot data.

RESPONSE: We thank Reviewer #4 for this comment. We have now provided the quantitation of western blot data as numbers below the panels to diminish the crowded appearance of including bar graphs. Please see **Fig. 3g, Supplementary Fig. 3g, h, Supplementary Fig. 4g, Fig. 5a, f and i.**

Reviewer #4 2 Interpretation of the data for why WT-bmal1 but not dHL-Bmal1 increase Sox9 expression in Bmal1-null YUMM2.1 cell (page 6 line 171-175) is not that convincing to me.

RESPONSE: We thank Reviewer #4 for this comment. We have now clarified our interpretation of the data but refrained from excessively surmising. With the loss of Bmal1, we found an association with reduced HIF-1 α protein expression. HIF-1 α has been reported to induce Sox9 expression (PMID 17913788). In the case of expression of WT Bmal1, we found a rescue of HIF-1 α protein expression that was not so robust with dHLH Bmal1 expression. As such, we surmise that the rescue of HIF-1 α in the context of loss of endogenous Bmal1 (ie, via CRISPR/Cas9 editing) correlates with Sox9 expression. Of course, this is a hypothetical scenario that requires future studies to fully delineate these correlations.

Reviewer #4 3 The results showed that WT-Bmal1 cell generated higher tumor volume than dHLH. Does that mean WT-Bmal1 has a higher Bmal1 protein level than dHLH? Why does dHLH-Bmal1 drives a more mesenchymal epigenetic state? Because it causes a stronger post-transcriptional mechanism?

RESPONSE: Our results indicate that WT-Bmal1 induced higher tumor volume than dHLH-Bmal1, but WT-Bmal1 protein level is not higher than dHLH-Bmal1. Please note that there may be some confusion about the notion of immunoresistance and overall tumor growth rates. It stands to reason that the overall tumor growth rate depends on: a) cell intrinsic proliferation rate (%Ki67 or tumor growth fraction), b) ability to resist cell death due to non-immune metabolic environment and c) ability to resist anti-tumor immunity. As such, dHLH confers immunoresistance as documented in our study using anti-PD1 treatment. The WT Bmal1 tumors, however, grew faster than dHLH tumors. We interpret these findings to mean that WT Bmal1 tumors have an intrinsic growth advantage along with immunoresistance, whereas dHLH tumors are highly immunoresistant but have decreased tumor cell

intrinsic growth in vivo. We surmise that the decrease in Sox10 expression, which is associated with melanoma tumor growth rate (PMID: 34879275), in dHLH Bmal1 tumors contributes to their slower cell intrinsic growth rate in vivo. We performed Ki67+ staining for melanoma cells in the tumor tissues and found that the percentage of Ki67+ melanoma cells in dHLH tumors are lower than both WT and EV tumors (new **Supplementary Figs. 2g and h**), which means a lower cell intrinsic growth rate.

Indeed, we believe that dHLH causes a stronger post-transcriptional effect through binding Myh9. We quantify the proximity labeling assay (PLA) and show in **revised Fig. 4g** and **Supplementary Fig. 4e** that dHLH Bmal1 quantitatively associates with Myh9 at a higher level than WT Bmal1.

Reviewer #4 Analytical approach
The analytical approach is correct.

RESPONSE: We thank Reviewer #4 for this comment.

Reviewer #4 Suggested improvements
The mechanism is clearly explained. Therefore, I don't suggest any more experiments.

RESPONSE: We thank Reviewer #4 for this comment. Please note that we have provided additional experimental data to address the issues raised by other reviewers.

Reviewer #4 Clarity and context
It would be better if the author provides some more background when it comes to the key words, for example, epithelial to mesenchymal transition and dHLH. That would be helpful for the readers to understand the whole picture.

RESPONSE: We thank Reviewer #4 for this comment. We now revised the manuscript to help make a complex story more approachable.

References
The references are appropriately cited.

RESPONSE: We thank Reviewer #4 for this comment.

REVIEWERS' COMMENTS

Reviewer #1 (Remarks to the Author):

The authors have done a nice job addressing my concerns, and also (in my opinion) the concerns of the other reviewers.

Reviewer #2 (Remarks to the Author):

The authors performed additional experiments to address some of my previous comments. The key issues regarding data interpretations related to Bmal1 dHLH over-expression and the discrepancy between the effects of loss of function and gain of function on HIF-1a activity remain unresolved. For the former, the authors argue that this reviewer was somewhat confused about the notion of immunoresistance and tumor promoting effect. If that's the case, they should address how dHLH promotes immunoresistance and inhibits tumor growth at the same time. Perhaps the RNAseq comparison between WT and dHLH Bmal1 over-expression, as mentioned in point 3 of my comments, will provide some hints.

While the manuscript might be complete in terms of molecular analysis, whether the various models make logical sense is unclear, to this reviewer. It's all too easy to use "context-dependent" when one can't explain the inconsistency in the experimental settings. This reviewer does recognize the authors' efforts in the revised manuscript. Perhaps at this stage it will be more productive to just agree to disagree.

Reviewer #3 (Remarks to the Author):

I thank the authors for providing a comprehensive rebuttal to my questions and criticism. I do not intend to reiterate the previous discussions, but there are still pending issues that, in my view, should need to be addressed:

1. The manuscript would benefit if the authors used the standard nomenclature for human and mouse genes throughout the manuscript. Sometimes, the authors refer Bmal1 as Arntl. Recently Arntl nomenclature was replaced by Bmal1 (see genecards website).

2. I strongly recommend the authors to provide the processed ChIPseq data as a supplemental table. After all, all processed data used to draw conclusions should be provided.

3. Importantly, the authors should provide a limitation statement to emphasize the strengths and weaknesses of the study. For example, the low sample size for several experiments. The conclusions of dHLH system have limitations, as stressed by other reviewers.

4. The authors stressed that no difference between BMAL1 protein between WT and dHLH cells is found. However, no quantification in any figure is provided in the revised version. Indeed, images from Fig 1g, 2a, 3g indicate higher BMAL1 expression (mutated form), but no quantification was provided. Notably, in the dHLH group, a different band pattern between WT and dHLH is observed. The authors should clarify this. Could one speculate that part of the dHLH effects is linked to a higher BMAL1 level than WT? The authors should discuss this further and clearly acknowledge further limitations.

Reviewer #4 (Remarks to the Author):

Reviewer #4 1 It would be easier to read if put a bar graph to show the quantity of protein levels of the western blot data.

RESPONSE: We thank Reviewer #4 for this comment. We have now provided the quantitation of western blot data as numbers below the panels to diminish the crowded appearance of including bar graphs. Please see Fig. 3g, Supplementary Fig. 3g, h, Supplementary Fig. 4g, Fig. 5a, f and i.

Comment: Thanks for adding the bar graphs.

Reviewer #4 2 Interpretation of the data for why WT-bmal1 but not dHLH-Bmal1 increase Sox9 expression in Bmal1-null YUMM2.1 cell (page 6 line 171-175) is not that convincing to me.

RESPONSE: We thank Reviewer #4 for this comment. We have now clarified our interpretation of the data but refrained from excessively surmising. With the loss of Bmal1, we found an association with reduced HIF-1 α protein expression. HIF-1 α has been reported to induce Sox9 expression (PMID 17913788). In the case of expression of WT Bmal1, we

found a rescue of HIF-1 α protein expression that was not so robust with dHLH Bmal1 expression. As such, we surmise that the rescue of HIF-1 α in the context of loss of endogenous Bmal1 (ie, via CRISPR/Cas9 editing) correlates with Sox9 expression. Of course, this is a hypothetical scenario that requires future studies to fully delineate these correlations.

Comment: I see that in figure1g that dHLH Bmal1 in Bmal1-null YUMM2.1 didn't rescue as much HIF-a protein as WT-Bmal1 did. Could you explain a bit more how this difference occurred?

Reviewer #4 3 The results showed that WT-Bmal1 cell generated higher tumor volume than dHLH. Does that mean WT-Bmal1 has a higher Bmal1 protein level than dHLH? Why does dHLH-Bmal1 drives a more mesenchymal epigenetic state? Because it causes a stronger post-transcriptional mechanism?

RESPONSE: Our results indicate that WT-Bmal1 induced higher tumor volume than dHLH-Bmal1, but WT-Bmal1 protein level is not higher than dHLH-Bmal1. Please note that there may be some confusion about the notion of immunoresistance and overall tumor growth rates. It stands to reason that the overall tumor growth rate depends on: a) cell intrinsic proliferation rate (%Ki67 or tumor growth fraction), b) ability to resist cell death due to non-immune metabolic environment and c) ability to resist anti-tumor immunity. As such, dHLH confers immunoresistance as documented in our study using antiPD1 treatment. The WT Bmal1 tumors, however, grew faster than dHLH tumors. We interpret these findings to mean that WT Bmal1 tumors have an intrinsic growth advantage along with immunoresistance, whereas dHLH tumors are highly immunoresistant but have decreased tumor cell intrinsic growth in vivo. We surmise that the decrease in Sox10 expression, which is associated with melanoma tumor growth rate (PMID: 34879275), in dHLH Bmal1 tumors contributes to their slower cell intrinsic growth rate in vivo. We performed Ki67+ staining for melanoma cells in the tumor tissues and found that the percentage of Ki67+ melanoma cells in dHLH tumors are lower than both WT and EV tumors (new Supplementary Figs. 2g and h), which means a lower cell intrinsic growth rate. Indeed, we believe that dHLH causes a stronger post-transcriptional effect through binding Myh9. We quantify the proximity labeling assay (PLA) and show in revised Fig. 4g and Supplementary Fig. 4e that dHLH Bmal1 quantitatively associates with Myh9 at a higher level than WT Bmal1.

Comment: Thanks for the explanation. I get the idea that the authors like to present. Given the complicated mechanism, I would suggest using a diagram to show the main idea of the paper if possible.

REVIEWERS' COMMENTS revision

Reviewer #1 (Remarks to the Author):

The authors have done a nice job addressing my concerns, and also (in my opinion) the concerns of the other reviewers.

RESPONSE: We appreciate the support of Reviewer #1, who assesses that we have addressed Reviewer #1 concerns as well as those of other reviewers.

Reviewer #2 (Remarks to the Author):

The authors performed additional experiments to address some of my previous comments. The key issues regarding data interpretations related to Bmal1 dHLH over-expression and the discrepancy between the effects of loss of function and gain of function on HIF-1 α activity remain unresolved. For the former, the authors argue that this reviewer was somewhat confused about the notion of immunoresistance and tumor promoting effect. If that's the case, they should address how dHLH promotes immunoresistance and inhibits tumor growth at the same time. Perhaps the RNAseq comparison between WT and dHLH Bmal1 over-expression, as mentioned in point 3 of my comments, will provide some hints.

RESPONSE: We regret that Reviewer #2 stipulates that dHLH promotes immunoresistance and “inhibits” tumor growth. In fact, dHLH over-expression modestly increased tumor volumes as compared with EV control transfected Yumm2.1 cells. As such, dHLH over-expression does not inhibit tumor growth per se. As indicated in the manuscript, we have not identified informative differences between WT and dHLH Bmal1 over-expression at the transcriptional level that fully explains the higher number of Ki67+ cells in WT tumors than dHLH tumors other than the more severe suppression of Sox10 by dHLH. Sox 10 expression level is associated with tumor growth as already cited in the manuscript. Please also note our previous response:

“The WT Bmal1 tumors, however, grew faster than dHLH tumors. We interpret these findings to mean that WT Bmal1 tumors have an intrinsic growth advantage along with immunoresistance, whereas dHLH tumors are highly immunoresistant but have decreased tumor cell intrinsic growth in vivo. We surmise that the decrease in Sox10 expression, which is associated with melanoma tumor growth rate (PMID: 34879275), in dHLH Bmal1 tumors contributes to their slower cell intrinsic growth rate in vivo.”

While the manuscript might be complete in terms of molecular analysis, whether the various models make logical sense is unclear, to this reviewer. It's all too easy to use "context-dependent" when one can't explain the inconsistency in the experimental settings. This reviewer does recognize the authors' efforts in the revised manuscript. Perhaps at this stage it will be more productive to just agree to disagree.

RESPONSE: We agree with Reviewer #2 that there appears to be some confusion over phenotypic changes due to Bmal1 loss-of-function versus Bmal1 over-expression. Indeed, our data indicates that over-expressing Bmal1 is not diametrically opposite of Bmal1 knockout. Hence, we used the term context-dependent. We appreciate that Reviewer #2 agrees to disagree.

Reviewer #3 (Remarks to the Author):

I thank the authors for providing a comprehensive rebuttal to my questions and criticism.

RESPONSE: We appreciate Reviewer #3 recognizing our comprehensive rebuttal.

I do not intend to reiterate the previous discussions, but there are still pending issues that, in my view, should need to be addressed:

- 1. The manuscript would benefit if the authors used the standard nomenclature for human and mouse genes throughout the manuscript. Sometimes, the authors refer Bmal1 as Arntl. Recently Arntl nomenclature was replaced by Bmal1 (see genecards website).*

RESPONSE: We thank Reviewer #3 for highlighting this, and we have accordingly edited the manuscript.

- 2. I strongly recommend the authors to provide the processed ChIPseq data as a supplemental table. After all, all processed data used to draw conclusions should be provided.*

RESPONSE: We now provide the ChIPseq data as supplementary Data with the further revised manuscript.

- 3. Importantly, the authors should provide a limitation statement to emphasize the strengths and weaknesses of the study. For example, the low sample size for several experiments. The conclusions of dhLH system have limitations, as stressed by other reviewers.*

RESPONSE: We referred to the limitation of the low sample size in the revised manuscript and now have further emphasized this in the limitations section at the end.

- 4. The authors stressed that no difference between BMAL1 protein between WT and dhLH cells is found. However, no quantification in any figure is provided in the revised version. Indeed, images from Fig 1g, 2a, 3g indicate higher BMAL1 expression (mutated form), but no quantification was provided. Notably, in the dhLH group, a different band pattern*

between WT and dHLH is observed. The authors should clarify this. Could one speculate that part of the dHLH effects is linked to a higher BMAL1 level than WT? The authors should discuss this further and clearly acknowledge further limitations.

RESPONSE: We showed that dHLH was in fact expressed slightly higher than WT Bmal1 by immunoblotting. Please see **Fig.3g** as a representative for Bmal1 protein quantification. In fact, the PLA quantitation also showed a higher association of dHLH Bmal1, compared to WT Bmal1, with Myh9. We have already alluded to this in our revised manuscript.

Reviewer #4 (Remarks to the Author):

Reviewer #4 1 *It would be easier to read if put a bar graph to show the quantity of protein levels of the western blot data.*

RESPONSE: We thank Reviewer #4 for this comment. We have now provided the quantitation of western blot data as numbers below the panels to diminish the crowded appearance of including bar graphs. Please see Fig. 3g, Supplementary Fig. 3g, h, Supplementary Fig. 4g, Fig. 5a, f and i.

Comment: Thanks for adding the bar graphs.

RESPONSE: We appreciate Reviewer #4 acknowledging our responsiveness to the suggestion.

Reviewer #4 2 *Interpretation of the data for why WT-bmal1 but not dHLH-Bmal1 increase Sox9 expression in Bmal1-null YUMM2.1 cell (page 6 line 171-175) is not that convincing to me.*

RESPONSE: We thank Reviewer #4 for this comment. We have now clarified our interpretation of the data but refrained from excessively surmising. With the loss of Bmal1, we found an association with reduced HIF-1 α protein expression. HIF-1 α has been reported to induce Sox9 expression (PMID 17913788). In the case of expression of WT Bmal1, we found a rescue of HIF-1 α protein expression that was not so robust with dHLH Bmal1 expression. As such, we surmise that the rescue of HIF-1 α in the context of loss of endogenous Bmal1 (ie, via CRISPR/Cas9 editing) correlates with Sox9 expression. Of course, this is a hypothetical scenario that requires future studies to fully delineate these correlations.

Comment: I see that in figure1g that dHLH Bmal1 in Bmal1-null YUMM2.1 didn't rescue as much HIF-a protein as WT-Bmal1 did. Could you explain a bit more how this difference occurred?

RESPONSE: We note that "dHLH Bmal1 in Bmal1-null YUMM2.1 didn't rescue as much HIF-a protein as WT-Bmal1 did." We believe, but do not wish to speculate in the manuscript, that WT-Bmal1 binds HIF-1a (PMID: 10864977) more effectively than dHLH-Bmal1, but this aspect is beyond the scope of the current experimental work.

Reviewer #4 3 *The results showed that WT-Bmal1 cell generated higher tumor volume than dHLH. Does that mean WT-Bmal1 has a higher Bmal1 protein level than dHLH? Why does dHLH-Bmal1 drives a more mesenchymal epigenetic state? Because it causes a stronger post-transcriptional mechanism?*

RESPONSE: Our results indicate that WT-Bmal1 induced higher tumor volume than dHLH-Bmal1, but WT-Bmal1 protein level is not higher than dHLH-Bmal1. Please note that there may be some confusion about the notion of immunoresistance and overall tumor growth rates. It stands to reason that the overall tumor growth rate depends on: a) cell intrinsic proliferation rate (%Ki67 or tumor growth fraction), b) ability to resist cell death due to non-immune metabolic environment and c) ability to resist anti-tumor immunity. As such, dHLH confers immunoresistance as documented in our study using antiPD1 treatment. The WT Bmal1 tumors, however, grew faster than dHLH tumors. We interpret these findings to mean that WT Bmal1 tumors have an intrinsic growth advantage along with immunoresistance, whereas dHLH tumors are highly immunoresistant but have decreased tumor cell intrinsic growth in vivo. We surmise that the decrease in Sox10 expression, which is associated with melanoma tumor growth rate (PMID: 34879275), in dHLH Bmal1 tumors contributes to their slower cell intrinsic growth rate in vivo. We performed Ki67+ staining for melanoma cells in the tumor tissues and found that the percentage of Ki67+ melanoma cells in dHLH tumors are lower than both WT and EV tumors (new Supplementary Figs. 2g and h), which means a lower cell intrinsic growth rate. Indeed, we believe that dHLH causes a stronger post-transcriptional effect through binding Myh9. We quantify the proximity labeling assay (PLA) and show in revised Fig. 4g and Supplementary Fig. 4e that dHLH Bmal1 quantitatively associates with Myh9 at a higher level than WT Bmal1.

Comment: Thanks for the explanation. I get the idea that the authors like to present. Given the complicated mechanism, I would suggest using a diagram to show the main idea of the paper if possible.

RESPONSE: We thank Reviewer #4 for appreciating our explanation. We believe that our key message is the unforeseen connection between Bmal1, Myh9, cell state and immune evasion. We request not providing an additional diagram to explain what could contribute to tumor growth volume.